# A growing bacterial colony in two dimensions as an active nematic

D. Dell'Arciprete [1,2], M.L. Blow[1], A.T. Brown[1], F.D.C. Farrell[1,3], J.S. Lintuvuori [4], A.F. McVey[1], D. Marenduzzo[1] & W.C.K. Poon[1]

How a single bacterium becomes a colony of many thousand cells is important in biomedicine and food safety. Much is known about the molecular and genetic bases of this process, but less about the underlying physical mechanisms. Here we study the growth of single-layer micro-colonies of rod-shaped *Escherichia coli* bacteria confined to just under the surface of soft agarose by a glass slide. Analysing this system as a liquid crystal, we find that growth-induced activity fragments the colony into microdomains of well-defined size, whilst the associated flow orients it tangentially at the boundary. Topological defect pairs with charges $\pm\frac{1}{2}$ are produced at a constant rate, with the $+\frac{1}{2}$ defects being propelled to the periphery. Theoretical modelling suggests that these phenomena have different physical origins from similar observations in other extensile active nematics, and a growing bacterial colony belongs to a new universality class, with features reminiscent of the expanding universe.

[1] SUPA, School of Physics and Astronomy, The University of Edinburgh, James Clerk Maxwell Building, Peter Guthrie Tait Road, Edinburgh EH9 3FD, UK. [2] Dipartimento di Fisica, Universita' di Roma La Sapienza, Piazzale Aldo Moro 5, 00185 Rome, Italy. [3] Life Sciences, University of Warwick, Coventry CV4 7AL, UK. [4] Université Bordeaux, CNRS, LOMA, UMR 5798, 33400 Talence, France. Correspondence and requests for materials should be addressed to D.M. (email: dmarendu@ph.ed.ac.uk)

Active matter occurs at many length-scales: from bird flocks[1], through shaken grains[2] and swimming bacteria[3] and synthetic colloids[4], to gels in which protein motors 'walk' on filamentous 'rails'[5]. All contain agents consuming energy to drive locomotion. The field attracts physicists (as a 'grand challenge' in non-equilibrium statistical mechanics), theoretical biologists (as a paradigm uniting phenomena across disparate scales) and materials scientists (who see applications in, e.g., self-assembly and artificial wound healing).

Active matter systems may show 'polar' ($2\pi$-periodic, like an arrow) or 'nematic' ($\pi$-periodic, like a double arrow) orientational ordering. Within the part of the system under consideration, momentum is either conserved or not. A bulk system which conserves momentum is termed 'wet', while in a 'dry' system, such as a two-dimensional (2D) layer interacting strongly with a substrate, momentum is not conserved. The polar/nematic and wet/dry dichotomies make up the four classical universality classes of active matter[6]. Recent developments show that 'wet' and 'dry' are the limiting cases of a continuum of possible theories[7], and that 'scalar active matter' exists with no orientational order[8]. In all these cases, the number of active agents is conserved at all times. Studies of biological morphogenesis as active matter have also, until recently[9,10], focussed on developmental phases with constant average particle number by balancing cell birth and death. In the following, we refer to these systems generically as 'number-conserving' active matter.

Here we instead study a simple active matter system with non-conserved particle number: a two-dimensional layer of dividing *Escherichia coli* cells confined in agarose gel by a glass slide. It represents the early stage of bacterial colonisation in many contexts, and constitutes perhaps the simplest example of biological morphogenesis. Despite the absence of cell differentiation, a growing bacterial colony is governed by similar physical constraints to those controlling eukaryotic morphogenesis, including tumour growth[11].

Cell-cell mechanical interactions constrain colony growth[12–19]. An elongating *E. coli* pushing against its neighbours in a colony has been modelled as an extensile dipole in a viscous liquid, i.e., a wet active nematic, in which viscous drag between the cells and bulk liquid controls colony morphology[10]. A dry continuum model has also been proposed to describe the chaotic behaviour of nematic domains in simulated colonies[19]. However, no theory of growing active nematics has yet been tested against quantitative experimental data.

We provide such data by analysing growing *E. coli* colonies as 'living anisotropic fluids'. We observe phenomena resembling those seen in number-conserving active nematics, including 'active anchoring' at the boundary[20] and the proliferation of topological defects[21]. However, the underlying mechanisms differ. We propose a theoretical framework in which our observations emerge from growth-driven Hubble-like expansive flows[22] and friction between cells and their substrate[17]. Alluding to this qualitative broad analogy, henceforth we call this class of systems (dry) 'Hubble active nematics'. [Clearly, this suggestive analogy comes with limitations. Most notably, within our quasi-2D colonies the amount of matter grows over time, where in the expanding universe it is the metric that changes in scale.]

## Results

**Morphology and growth rate**. We begin by reporting results on the morphology of our growing microcolonies (see Methods for experimental details on growth conditions and image processing). Confocal imaging (Supplementary Fig. 1) shows that cells are embedded just underneath the agarose surface, confined on one side by the cover slip. Figure 1 shows snapshots from a typical colony, which are representative of the behaviour we observe under our experimental conditions (see Methods). A number of qualitative changes occur as cells lengthen and divide. At early times, the colony is elongated, Fig. 1a, b, as cells grow longitudinally along a common axis. Later, this 1D order breaks down, giving way to a pattern of nematic domains bordered by defects, Fig. 1c. During this stage, cells push outwards in all directions, and the colony becomes more isotropic, Fig. 1 (inset, see also Supplementary Note 1 and Supplementary Fig. 2). Under our conditions, the middle of the colony turns phase dark after ~5 h when it contains $N \lesssim 10^3$ cells, Fig. 1d (dark spots), indicating buckling of the 2D layer into the third dimension[17], and we stop our analysis. Typically, buckling occurs when the colony size $2R \gtrsim$ 50 µm. All colonies show the same trend, although quantitative details are subject to noise (Supplementary Fig. 2).

Cells in the middle of a large colony may grow slower due to nutrient limitation. Measurement of cell lengths as a function of time, Fig. 2a, shows that their growth rate is in fact spatially uniform to within experimental uncertainties, so that nutrient limitation is unimportant for our colonies (although the growth rate is time dependent, Supplementary Note 2 and Supplementary Fig. 2b). The near-uniform growth rate is also the reason why we do not observe fingering instabilities, which appear when growth rate is limited by nutrient[16] or depends on local density[23].

**'Hubble' expansion**. We found no coherent azimuthal movement of cells, but there is a coherent radial movement, whose speed increases linearly with distance from the centroid, Fig. 2b, reminiscent of the Hubble expansion of the universe. Consider the velocity field, **v**, of this growth-induced motion: we now write an equation for this quantity based on our experimental observations. Mass conservation requires that the cell density, $\rho$, satisfies

$$\partial_t \rho + \nabla \cdot (\rho \mathbf{v}) = D_t \rho + \rho \nabla \cdot \mathbf{v} = \Lambda \rho, \qquad (1)$$

where $\Lambda$ is the growth rate, and $D_t = \partial_t + \mathbf{v} \cdot \nabla$ is the material derivative. Assuming incompressibility (we relax this assumption below), i.e., $D_t \rho = 0$, we find

$$\nabla \cdot \mathbf{v} \equiv \partial_i v_i = \Lambda, \qquad (2)$$

where hereafter repeated suffices are summed. Since $\Lambda$ is independent of space in our colonies, and there is no coherent azimuthal movement, Eq. (2) predicts the radial component to be $v_r = \Lambda r/2 \equiv Hr$, as observed, Fig. 2b (the factor 1/2 arises as the system is 2D).

A linear fit to the population-averaged data in Fig. 2b gives our 'Hubble constant' $H = 0.007\,\text{min}^{-1}$. Theoretically, $H = 0.5\Lambda$, while individual values of $(H, \Lambda)$ obtained from each of the 32 colonies, Fig. 2c, are consistent with a linear scaling of $H$ with $\Lambda$ and are fitted by $H = 0.4\Lambda$ (the correlation between $H$ and $\Lambda$ is $r \sim 0.77$, corresponding to a $p$-value $p < 0.00001$). The discrepancy from the theoretical prediction $H = 0.5\Lambda$ may arise either from small systematic errors in the fitting (e.g., due to small drift close to colony centroids) or from the approximations in the theoretical argument, such as that of isotropic growth. The scatter in the data in Fig. 2c gives an indication of the extent of intrinsic noise in our population of colonies.

Note that the expansive 'Hubble flow' we observe is distinct from the spontaneous shear flow exhibited by number-conserving active nematics[24,25].

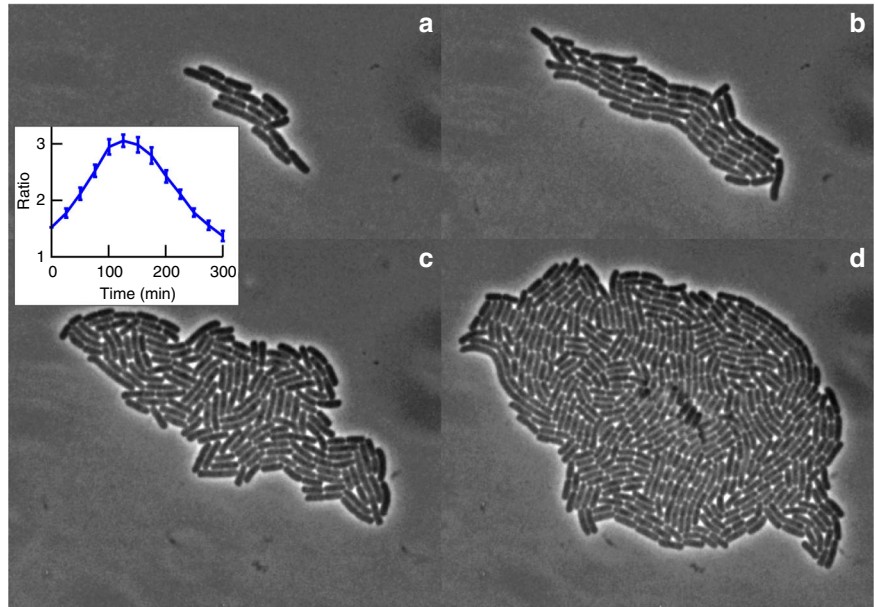

**Fig. 1** Morphology of a growing bacterial colony. Snapshots of a growing colony, from a few cells (**a**; $t = 125$ min, $N = 15$), through the early stages of growth where the colony elongates (**b**; $t = 171$ min, $N = 56$), followed by later stages of growth in which the colony shape becomes less anisotropic (**c**; $t = 216$ min, $N = 164$), up to the time immediately prior to the colony invading the third dimension (**d**; $t = 261$ min, $N = 475$). Inset: The average ratio of long and short axes of 32 colonies plotted against time. The axes are estimated as the square roots of the larger and smaller eigenvalue of the second moment of area. Error bars are standard errors of the mean

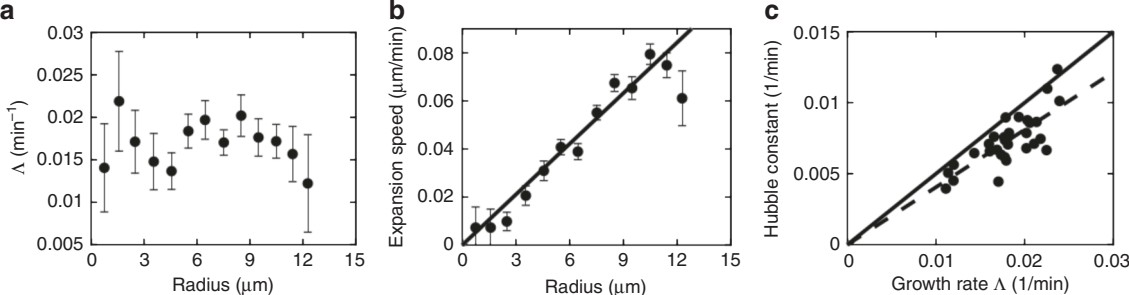

**Fig. 2** 'Hubble' cellular flow. **a** Plot of azimuthally-averaged growth rate against distance from the colony centroid, averaged over 32 colonies 10 min prior to buckling. The growth rate is approximately constant spatially. Error bars are standard errors of the mean. **b** Plot of the azimuthually-averaged radial velocity of bacteria as a function of distance from the centroid of the colony, averaged over all colonies 10 min prior to buckling (see Supplementary Note 3 for details). Solid line: best fit. Error bars are standard errors of the mean. **c** Plot of the 'Hubble constant' for the individual colonies, plotted against the bacterial growth rate. Dashed line: best fit; solid line: $H = 0.5\Lambda$

**Active anchoring**. A striking feature of the larger colonies is that cells tend to be aligned tangentially at the periphery, Fig. 1d. We fitted a Bézier curve through the centroids of the outermost cells, and measured the angle $\psi$ between the curve and each boundary cell. For a typical colony just before it buckled, most peripheral cells show $\psi \lesssim 30°$, Fig. 3a. Such 'active anchoring' is also seen in simulated number-conserving active nematics[20], where it originates from the ingress of topological defects following the buckling of the boundary between an active nematic and an isotropic continuum. We will see later that the mechanism in our case is different.

**Continuum description**. Turning to the bulk of our colonies, we first construct a continuum structural description based on our raw data, which consist of the position, length and orientation of each cell as functions of time, $\{\mathbf{r}^i(t), L_i(t), \mathbf{v}^i(t)\}$. We start this coarse graining procedure by defining a function that 'smears out'

individual rod-shaped bacteria of diameter $W$:

$$f^i(\mathbf{r}) = \frac{1}{2}\left(\tanh\left[\frac{\mathbf{v}\cdot\Delta\mathbf{r}^i + \frac{1}{2}L^i}{\sigma}\right] - \tanh\left[\frac{\mathbf{v}\cdot\Delta\mathbf{r}^i - \frac{1}{2}L^i}{\sigma}\right]\right)$$
$$\times \frac{1}{2}\left(\tanh\left[\frac{\bar{\mathbf{v}}\cdot\Delta\mathbf{r}^i + \frac{1}{2}W}{\sigma}\right] - \tanh\left[\frac{\bar{\mathbf{v}}\cdot\Delta\mathbf{r}^i - \frac{1}{2}W}{\sigma}\right]\right), \quad (3)$$

with $\Delta\mathbf{r}^i = \mathbf{r} - \mathbf{r}^i$, $\bar{\mathbf{v}}^i$ the unit vector perpendicular to $\mathbf{v}^i$ and $\sigma$ a smoothing length. We use $\sigma = W$ to probe ordering at the single-cell level ($\sigma \to 0$ gives sharp $L_i \times W$ rectangles). The integrated cell 'mass' is preserved as $\sigma$ varies. [Note that the functional form of $f^i$ we use is required to account for the non-spherical, rod-like shape of bacteria.] The (number) density field is then

$$\rho(\mathbf{r}) = \sum_{i=1}^{N} f^i(\mathbf{r}). \quad (4)$$

To quantify orientational order, we assume that the two poles of a cell are indistinguishable (i.e., $\pm\boldsymbol{\nu}^i$ are equivalent), and construct a traceless and symmetric Q-tensor[26] field

$$Q_{\alpha\beta}(\mathbf{r}) = \sum_{i=1}^{N} f^i(\mathbf{r}) \left( 2\nu_\alpha^i \nu_\beta^i - \delta_{\alpha\beta} \right), \qquad (5)$$

$$= S(\mathbf{r}) \left[ 2n_\alpha(\mathbf{r})n_\beta(\mathbf{r}) - \delta_{\alpha\beta} \right], \qquad (6)$$

where we have also defined the scalar order parameter

$$S(\mathbf{r}) = \sqrt{Q_{xx}^2(\mathbf{r}) + Q_{xy}^2(\mathbf{r})}, \qquad (7)$$

and the director $\mathbf{n}(\mathbf{r})$, which is the unit eigenvector associated with the positive eigenvalue of $Q_{\alpha\beta}(r)$. Supplementary Fig. 3 (see Supplementary Note 4) shows $Q_{\alpha\beta}(\mathbf{r})$ of the snapshots in Fig. 1.

**Global order**. The global version of Eq. (7) quantifies the degree of orientational order averaged over an N-cell colony:

$$S(N) = \sqrt{\langle Q_{xx}\rangle^2 + \langle Q_{xy}\rangle^2} = \frac{1}{N}\sqrt{\sum_{i,j} \cos\left[2\left(\theta^i - \theta^j\right)\right]}, \quad (8)$$

where $\theta^i$ is the angle between the axis of cell $i$ and the director (i.e., $\boldsymbol{\nu}^i \times \mathbf{n} = \cos\theta^i$), and $\langle\ldots\rangle$ denotes averaging over a colony. This evaluates to unity when all cells are perfectly aligned, and to $1/\sqrt{N}$ for total orientational disorder. The measured degree of global order, $S(N)$, varies significantly between colonies, Fig. 3b, but the trend is clear. Global order is high for up to $\gtrsim 3$ generations ($N \lesssim 10$), and then decreases. Significantly, $\langle S\rangle = 1/\sqrt{N}$ forms the lower envelope of the vast majority of our data. Thus, while there is no sustained growth-induced global ordering[14], substantial residual order persists up to the buckling point, when our observations stop.

**Correlation lengths and domains**. Underlying the persistence of residual global order is a significant degree of local orientational order in the form of domains of aligned cells: Fig. 1c, d. Such local order can be quantified by calculating two-point correlation functions of the Q-tensor, $C^{\parallel}(r)$ and $C^{\perp}(r)$ (Supplementary Note 5), which measure the azimuthally-averaged orientational correlation at a distance $r$ from a typical cell parallel and perpendicular to its axis, respectively. These functions for the colony snapshots in Fig. 1 are shown in Supplementary Fig. 4. In the earliest stage of growth ($N = 15$), $C^{\parallel}(r)$ and $C^{\perp}(r)$ decay over $\approx 10$ and $5\,\mu m$ respectively, i.e., the whole colony is ordered (as is obvious from Fig. 1a). At intermediate times, parallel correlations grow faster than perpendicular correlations, before $C^{\parallel}(r)$ and $C^{\perp}(r)$ become more comparable at late times.

This trend is generic. We calculated $C^{\parallel}(r)$ and $C^{\perp}(r)$ for 32 colonies, and obtained correlation lengths defined by

$$l_c^{\parallel,\perp} = \frac{1}{C^{\parallel,\perp}(0)} \int_0^\infty C^{\parallel,\perp}(r)\mathrm{d}r, \qquad (9)$$

so that two cells each of length and width $L \gtrsim 2\,\mu m$ and $W \lesssim 1\,\mu m$ placed side by side and end-on give $l_c^{\parallel} = L/2 \approx 1\,\mu m$ and $l_c^{\perp} = W/2 \approx 0.5\,\mu m$ respectively. At early times, $l_c^{\parallel}/l_c^{\perp}$ grows (Supplementary Fig. 5), with $l_c^{\parallel}$ growing faster and peaking at $\lesssim 5\,\mu m$, compared to a maximum $l_c^{\perp} \lesssim 3\,\mu m$, corresponding to domains of some tens of cells, Fig. 1b. Thereafter, both correlation lengths decrease to $\gtrsim 2\,\mu m$, corresponding to domains of $\lesssim 10$ cells. Similarly-sized 'micro-domains' have been seen in recent simulated 2D bacterial colonies at late times[19].

To highlight the role of active growth in generating this pattern of behaviour, we performed Monte Carlo simulations[27–31] of passive (non-growing) sphero-cylinders[32] with lengths and widths taken from bacterial colonies (see Supplementary Note 6 and Supplementary Fig. 6). In the passive case, $l_c^{\parallel}/l_c^{\perp} \approx 1$ throughout the entire history of the colony (Supplementary Fig. 5).

**Proliferation of topological defects**. The existence of domains implies topological defects, where the director field is ill-defined. Defects control the elasticity and dynamics of passive liquid crystals, and feature prominently in number-conserving active nematics[6], where they arise from the self-sustaining shear flow arising from the stirring of the fluid by active particles.

The topological charge, or winding number, of a defect, is the accumulated change (in units of $2\pi$) in the angle $\phi$ made by the director field relative to an arbitrary direction on following a closed loop, $\mathcal{C}$, around the defect: $m = \frac{1}{2\pi}\oint_\mathcal{C} d\phi$. Defects are identified via a topological charge density[20]:

$$q = \frac{1}{2\pi}\left(\partial_x Q_{xx}\partial_y Q_{xy} - \partial_x Q_{xy}\partial_y Q_{xx}\right), \qquad (10)$$

which can be computed directly from the Q-tensor field.

An example of spatial maps of $q$ is shown in Supplementary Fig. 7 (see also Supplementary Note 7). We identified individual defects as local maxima and minima of $q(\mathbf{r})$, and determined their polarity from the unit vector $\hat{p}_{d,\alpha} = -\partial_\beta Q_{\alpha\beta}/\left|\partial_\beta Q_{\alpha\beta}\right|$ at the local maximum. As in conventional active nematics, defects in our colonies mostly have topological charge $\pm\frac{1}{2}$. As the colony grows, we see the formation of multiple defect pairs in the colony interior, Fig. 4a. While defects are created continuously, the total signed topological charge $\int q\mathrm{d}^2\mathbf{r} = 0$. However, the total number of defects, which we compute as the total unsigned topological charge, $\int|q|\mathrm{d}^2\mathbf{r}$ scales near-linearly with $N$, Fig. 4b, suggesting that defect generation and annihilation is a bulk, not boundary, phenomenon, and that topological (unsigned) charge grows exponentially in time. Linear scaling with $N$ is expected on general grounds if, as we observe, the colony fragments into micro-domain of well-defined size ($\lesssim 10$ cells).

**Defect dynamics**. Tracking the topological defects through time, Fig. 4a (see also Supplementary Fig. 7), shows that the orientation of the three-fold symmetric $-\frac{1}{2}$ defects does not significantly impact their dynamics[21]. Presumably, these are simply advected by the expansion of the colony, perhaps aided by mutual elastic interactions. In contrast, comet-like 'head and tail' $+\frac{1}{2}$ defects show marked directed motion—the speed just prior to buckling is $v_+ \sim 0.05\,\mu m\,s^{-1}$. [This value should be seen as an order-of-magnitude estimate as the measurement requires unambiguous tracking of defects which we could only do in few selected cases.] The distribution of $\hat{p}_d \cdot \Delta\hat{r}_{d-com}$, Fig. 4c, where $\Delta\hat{r}_{d-com}$ is the unit vector pointing from a colony's centre of mass to the defect, shows that positive defects propel along the direction of $\hat{p}_d$ (i.e., tail to head), reminiscent of their behaviour in a non-growing extensile active nematic[21].

The spatial distribution of $\pm\frac{1}{2}$ defects, Fig. 4d, shows 'charge separation'. There is an excess of $-\frac{1}{2}$ defects near the colony centre, while $+\frac{1}{2}$ defects predominate at the periphery. Presumably, inwardly-directed $+\frac{1}{2}$ defects have a high chance of annihilating $-\frac{1}{2}$ defects advected by colony expansion, while outward-directed $+\frac{1}{2}$ defects rapidly propel to the colony edge, where advected $-\frac{1}{2}$ defects have not yet reached. Moreover, once a $+\frac{1}{2}$ defect has propelled itself to the colony edge in a tail-to-head

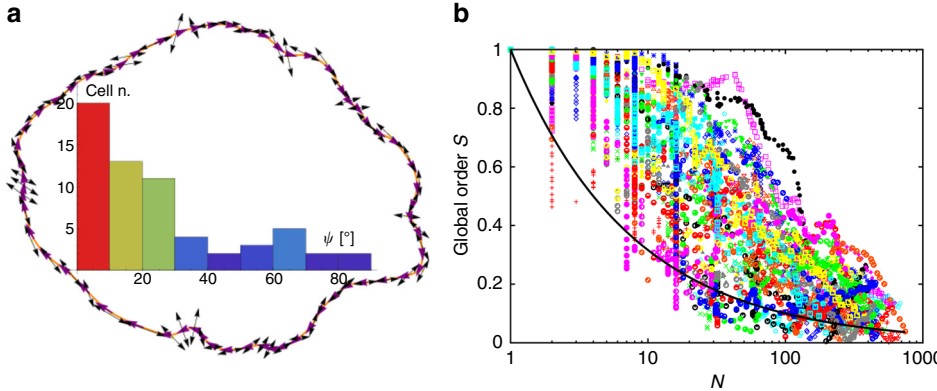

**Fig. 3** Active anchoring and global order. **a** Bézier curve constructed through the centroids of the bacteria at the boundary of a typical colony. Inside, the histogram of the (acute) angle $\psi^j$ between the boundary tangent and bacteria directors $\nu$ (black arrows). Most cells tend to align with the tangent of the colony boundary. **b** Plot of the global order, $S$, as a function of colony size for 32 colonies. The continuous curve is $S = 1/\sqrt{N}$

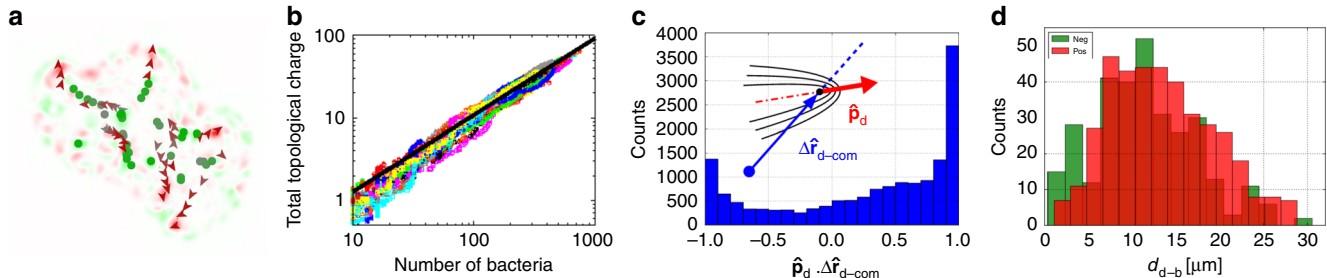

**Fig. 4** Topological defects in growing bacterial colonies. **a** Trajectory of topological defects within one colony. The $+1/2$ defects are shown in red, with the arrows denoting polarity, while the $-1/2$ defects are shown in green. The markers are increasingly discoloured for earlier times. **b** Plot of the total unsigned topological charge $\int |q| d^2\mathbf{r}$ versus instantaneous number of bacteria in the colony (data for all 32 colonies are included). Solid line, fit: $0.14N^{0.93}$, showing that this quantity increases approximately linearly with $N$. **c** Histogram of $\hat{\mathbf{p}}_d \cdot \Delta\hat{\mathbf{r}}_{d-com}$ for all $+1/2$ defects. When this quantity is positive, the corresponding $+1/2$ defect is directed outwards with respect to the centre of the colony. [For randomly oriented defects we expect a distribution peaked at $\pm 1$, and $\sim 1/\sqrt{1 - (\hat{\mathbf{p}}_d \cdot \Delta\hat{\mathbf{r}}_{d-com})^2}$.] **d** Spatial distribution of $\pm\frac{1}{2}$ defects as a function of their distance from the buckling point, $d_{d-b}$, showing evidence for charge separation, with an excess of $-\frac{1}{2}$ defects (green) near $d_{d-b} = 0$

direction (Fig. 4c), the director field pattern near its 'head' matches the tangential anchoring, Fig. 3a, and is therefore energetically favoured. Thus, we find that $+\frac{1}{2}$ defects are rarely ejected from the boundary into the colony. This contrasts with a number-conserving active nematic surrounded by an isotropic medium, where ejection of $+\frac{1}{2}$ defects from a deeply-undulating boundary into the interior is prominent[20].

Intriguingly, the region in which we observe an excess of $-\frac{1}{2}$ defects, $r \lesssim 0.2R$ (here $R$ is the radius of the inscribed circle), is precisely where we see the onset of buckling into the third dimension. This is reminiscent of the physics of passive liquid crystals, where the director field can avoid the formation of a costly defect core by escaping into the third dimension. This observation also suggests that defects may provide a means to relax stress in the system, as proposed recently for eukaryotic colonies[33].

**Theoretical description and length scales.** Our observations and analyses so far suggest that it should be fruitful to describe a growing 2D colony of *E. coli* cells theoretically as an active nematic, albeit in a different universality class than almost all current theories of this kind because of particle number non-conservation. Here, we first delineate a number of components that must feature in such a theory and point out their implications. We then perform numerical simulations of a set of equations which assumes a particularly simple constitutive relation for the active stress.

There are two important length scales in our system. The first one, $l_a$, is related to the onset of the 'generic instability' in active nematics. In wet systems, $l_a$ is given by a balance between active and elastic stresses[34], as $l_a \sim \sqrt{K/a}$, where $K$ is a Frank elastic constant and $a$ is the active stress coefficient, defined so that the growth-dependent active stress scales as $aQ_{\alpha\beta}$. In a growing colony, $K$ and $a$ both depend on growth-mediated cell deformations, and should scale as the cell wall elastic modulus, $G$, so that $a \sim G$ and $K \sim GL^2$ for cells of length $L$. They should also scale similarly with cell density[16,19], so that we may expect $K/a$ to be a constant, and $l_a \sim L$. In our dry system, the functional dependence of $l_a$ on parameters is more complicated (see below and[35]). However, as $l_a$ is generally lower for the dry than for the wet limit[7], we expect $l_a \sim L$ should still hold for our colonies.

A growing cell pushes and deforms its neighbours, generating stresses that scale as $G$. The resulting cell movements are opposed by dissipative forces, which we take to be lubricated frictional sliding between cells and agarose or glass controlled by a frictional drag coefficient per unit length, $\gamma$, so that the drag force on a cell of length $L$ sliding at speed $v$ is $\sim \gamma L v$. Force balance in terms of the stress tensor $\sigma_{\alpha\beta}$ then reads

$$\partial_\beta \sigma_{\alpha\beta} = (\gamma L \rho) v_\alpha \equiv \hat{\gamma} v_\alpha, \tag{11}$$

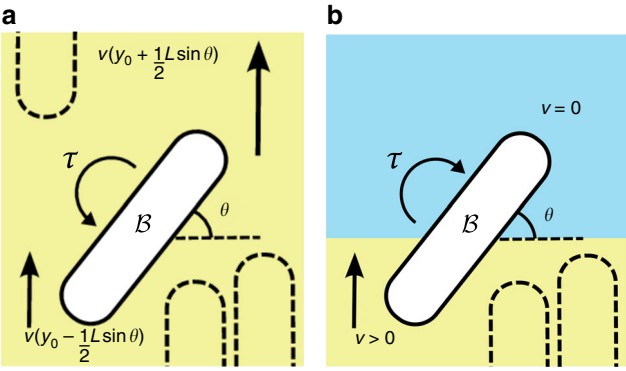

**Fig. 5** Growth-induced ordering. **a** Schematic illustration of a bacterium $\mathcal{B}$ in the bulk of colony, with the surrounding bacteria orientated to the vertical as exemplified by the dotted shapes. The growth-induced velocity gradients in the surrounding bulk produce a torque on $\mathcal{B}$, which rotates it into the same orientation as its neighbours. **b** A bacterium at the boundary of the colony experiences a steep velocity gradient of the opposite direction with respect to the case in **a**: the associated cellular flow rotates $\mathcal{B}$ into a tangential orientation to the boundary

where $\rho$ is the cell density. Our 'dry' description in Eq. (11) is motivated by the fact that our colony is very close to the glass coverslip (Supplementary Fig. 1); in general this equation will hold above a hydrodynamic 'screening length' $l_s \sim \sqrt{\frac{\eta}{\hat{\gamma}}}$, with $\eta$ the effective viscosity of the colony.

Growth stress propagates by diffusion; dimensional analysis gives the diffusivity $\sim G/\hat{\gamma}$[16]. [More rigorously, we can show that, within our theory, when $\Lambda = 0$ both $\rho$ and the stress evolve according to a diffusion equation where the diffusion coefficient is $\sim G/\hat{\gamma}$, given the simple constitutive relation for the stress considered below.] The extent of stress propagation during the lifetime of a single cell (birth to division) defines a stress propagation length

$$l_\sigma = \sqrt{\frac{G1}{\hat{\gamma}\Lambda}}. \tag{12}$$

Previous measurements give $G \sim 10^6$ Pa, albeit with large variations, while we estimate $\hat{\gamma} \sim 10^{15}$ (Pa s)m$^{-2}$ using previous experimental results[36] (Supplementary Note 8). Thus $l_\sigma \sim$ mm. This is an upper bound for the size of a colony in which internal growth stresses can relax into the agarose matrix. In practice, buckling into the third dimension will occur when the stress approaches the yield stress of 2% agarose, $\sim 3 \times 10^3$ Pa $\ll G$. With this revised stress estimate, $l_\sigma \sim 50$ μm, similar to experiments. Intriguingly, $v_+\Lambda^{-1} \sim 50$ μm as well, suggesting that defects may have a role in relaxing growth stresses.

**Growth-induced alignment**. To see how growth should induce orientational order, consider first a system of bacteria oriented along $y$ ($Q_{yy} = 1$, $Q_{xx} = -1$, $Q_{xy} = Q_{yx} = 0$), Fig. 5a, with a single slightly-misaligned bacterium $\mathcal{B}$ (centroid at $y_0$) whose angle $\theta$ with the orthogonal ($x$) axis is <90°. The only velocity gradient component is $\partial_y v_y = \Lambda$. The fraction of $\mathcal{B}$ at larger $y$ moves with higher $v_y$, and so must experience a higher force than the lower part; the net torque realigns this bacterium.

Quantitatively, the force giving rise to the velocity of a segment, $dl$, of $\mathcal{B}$ located a distance $l$ from the centroid must balance the frictional drag with the substrate at this position, $\gamma v_y(y)dl$. The

total torque on $\mathcal{B}$ is then

$$\tau = \int_{-L/2}^{L/2} \gamma v(y_0 + l\sin\theta) l\cos\theta dl. \tag{13}$$

Setting $\tau = \zeta d\theta/dt$, where $\zeta$ is the rotational drag on a single cell due to its neighbours, and expanding to first order gives

$$\partial_t \theta = \frac{\gamma L^3}{24\zeta} \left(\partial_y v_y\right) \sin 2\theta, \text{[1D]}, \tag{14}$$

where $\partial_y v_y = \Lambda > 0$, Eq. (14) has a stable equilibrium at $\theta = 90°$, i.e., growth drives alignment.

This quasi-1D analysis correctly predicts a high degree of alignment in our early-stage colonies, Fig. 1a, b. This persists only for a few generations, because the active instability length scale $l_a \sim L$. When the colony size $2R$ increases to the point when $2R/l_a \gg 1$, the extensile stresses due to growth create bend distortions, as in number-conserving active nematics. The colony now grows in two dimensions.

For simplicity, consider the simplest case of circularly symmetric expansion in 2D, for which the cell velocity field is

$$\mathbf{v} = g(r)\mathbf{r}, \tag{15}$$

with $g(r)$ a local growth rate. Substituting Eq. (15) into nematohydrodynamic equations describing flow-orientation coupling (see Supplementary Note 9) yields the analogue of Eq. (14) for 2D:

$$\partial_t \theta = \frac{\xi g' r}{4S} \sin 2(\theta - \phi), \text{[2D]} \tag{16}$$

where $\phi$ denotes the azimuthal angle in polar coordinates and $\xi > 0$ is a flow-aligning parameter.

Equation (16) predicts that growth-induced alignment depends on the sign of $g'$. For compressible flows, $g' > 0$, there is a stable steady state ($\partial_t\theta = 0$) with $\theta = \phi$, giving a radially-ordered colony with a central $+1$ vortex defect. This is the analogue of the growth-induced alignment predicted by Eq. (14) for 1D.

We do not observe such alignment, because experimentally, we find incompressibility, i.e., $g \simeq \Lambda/2$ and $g' = 0$, Fig. 2a. In this case, the steady state of Eq. (16) does not require any particular relation between $\theta$ and $\phi$. The colony breaks into domains with a length scale controlled by $l_a$. The number of defects scales as the number of domains. There are $\sim 10 \approx 3^2$ cells per defect, Fig. 4b, consistent with $l_a \sim L$. Our colonies are effectively incompressible because their sizes ($2R \lesssim 50$ μm) are always $< l_\sigma$, so that growth-induced stresses relax quickly.

At the edge of a colony, $g \simeq \Lambda/2$ (inside) transitions to $g = 0$ (outside), so that $g' < 0$. Equation (16) predicts a stable steady state with $\theta = \phi + \pi/2$, i.e., tangential alignment, again as observed. Qualitatively, the origins of this effect is easy to understand. The presence and absence of growth-generated forces inside and outside the colony respectively, Fig. 5b, means that a cell not aligned tangentially will be rotated back into such alignment, as previously suggested[37].

**Non-uniform active stress**. In theories of number-conserving active nematics, the active stress is taken to be

$$\sigma_{\alpha\beta}^{(a)} = -aQ_{\alpha\beta}, \tag{17}$$

where $a$ is a positive or negative constant for extensile or contractile activity respectively[6]. It is tempting to model growing bacterial colonies in the same way with $a \propto \Lambda$.

To see why this does not work for a Hubble active nematic, consider again a 1D colony in the $y$ direction. Force balance, Eq.

(11), and mass conservation, Eq. (1), give

$$\partial_{yy}\sigma_{yy}^{(a)} = \hat{\gamma}\Lambda. \quad [1D] \qquad (18)$$

For a 1D colony occupying $0 < y < y_0$ with spatially-constant $\Lambda$ (cf. Fig. 2a), Eq. (18) solves to[14,15]

$$\sigma_{yy} = -P_0 - \tfrac{1}{2}\Lambda\hat{\gamma}(y_0^2 - y^2), \qquad (19)$$

where $P_0$ is the pressure exerted by the agarose at the boundaries. The active stress field is non-uniform, increasing from the periphery to the centre by $\Delta\sigma \sim \hat{\gamma}\Lambda R^2$ for a colony of dimension $2R$. Numerically, this inhomogeneity can be substantial (our estimated parameters give $\Delta\sigma \sim 3 \times 10^3$ Pa for $2R \sim l_\sigma$). Physically, such non-uniformity necessarily arises because peripheral cells have to push at fewer cells sliding frictionally against the substrate as the colony expands[17].

The coupling of mechanics and nematic order in 2D gives rise to new complexity. Inspired by Eq. (19), we may expect

$$\sigma_{\alpha\beta} = -p(\mathbf{r})\delta_{\alpha\beta} - a(\mathbf{r})Q_{\alpha\beta}. \quad [2D] \qquad (20)$$

A simple constitutive relation consistent with a dry compressible active nematic is $p = G\max[(\rho/\rho_0 - 1), 0]$, and $a = a_0\rho$, where $\rho_0$ is the close packed density of undeformed cells. The equations resulting from substituting this or other constitutive relations into the laws of nematohydrodynamics are only tractable by numerical simulations (see below). Nevertheless, the same physics we appealed to in the 1D case will still give rise to a stress field increasing from periphery to centre.

In a sufficiently small colony, $2R \ll l_\sigma$, these non-uniform growth-induced stresses relax rapidly through the periphery into the agarose, so that the flow is effectively incompressible, Fig. 2b. As $2R \to l_\sigma$, however, stress relaxation becomes progressively more incomplete, eventually leading to the buckling when the colony is not confined to 2 dimensions (as is the case in our experiments).

**Active field theory simulations.** To complete our theoretical description, we now report results from direct numerical simulations of a full model comprising Eqs. (1) and (11), together with the following evolution equation for the liquid crystal order parameter,

$$\begin{aligned}\frac{\partial Q_{\alpha\beta}}{\partial t} + v_\gamma\partial_\gamma Q_{\alpha\beta} = &\ \hat{K}\partial_\gamma\partial_\gamma Q_{\alpha\beta} + \alpha(\rho)Q_{\alpha\beta} \\ &- \beta Q_{\gamma\delta}^2 Q_{\alpha\beta} + \xi\left(u_{\alpha\beta} - u_{\gamma\gamma}\frac{\delta_{\alpha\beta}}{2}\right) \\ &+ \omega_{\alpha\gamma}Q_{\gamma\beta} - Q_{\alpha\gamma}\omega_{\gamma\beta}.\end{aligned} \qquad (21)$$

In Eq. (21), we denoted $\hat{K} = K/\gamma_1$, with $\gamma_1$ the rotational viscosity of the liquid crystalline colony, whereas $u_{\alpha\beta} = (\partial_\beta v_\alpha + \partial_\alpha v_\beta)/2$ and $\omega_{\alpha\beta} = (\partial_\beta v_\alpha - \partial_\alpha v_\beta)/2$. We also introduced $\alpha(\rho) = \alpha_0(\rho - \rho_c)$, with $\alpha_0 > 0$ and $\rho_c \equiv \rho_0/2$ to describe liquid crystalline ordering à-la-Onsager for sufficiently large bacterial density—the term proportional to $\beta$ is a saturation term, and we choose $\beta = \alpha_0\rho_0/2$ which yields $2S^2 = 1$ in the middle of the colony. The active stress in Eq. (11) is given by Eq. (20), with $p = G\max[(\rho/\rho_0 - 1), 0]$, and $a = a_0\rho$. For other parameter values, see Fig. 6; for more details, see Methods and Supplementary Figs. 8 and 9.

Our simulations confirm all qualitative predictions of our theory, and also compares favourably with experiments, see Fig. 6 and Supplementary Fig. 8. First, as expected the radial velocity increases linearly with distance to the colony centre—interestingly, our simulations also typically show a slightly smaller 'Hubble constant' than expected, as in experiments ($H < \Lambda/2$, Figs. 2b and 6a). Second, simulations reproduce the tangential alignment at the colony boundaries (Fig. 6b and Supplementary Fig. 8). Third, when the active coefficient $a_0$ is sufficiently large, the colony breaks up into domains of well-defined lengthscale, $l_a$, separated by defects (Fig. 6b and Supplementary Fig. 8). Fourth, the total topological charge increases near-linearly with colony area (proportional to number of bacteria, see Fig. 6c). Finally, the positive topological charges (mainly associated with $+1/2$ defects) are more peripheral than the negative ones (mainly associated with $-1/2$ defects, Fig. 6d).

An analysis of our data suggest that there is a substantial range of $a_0$ for which $l_a \sim a_0^{-1/2}$ (Supplementary Fig. 9). This is in line with simulations of turbulent-like dynamics in number-conserving dry active nematics[35]. Comparison with the latter study and with the results of[19], together with dimensional analysis, suggest that for large $a_0$ a possible functional form for the domain size may be $l_a \sim \sqrt{\frac{\hat{K}\hat{\gamma}}{a_0}}\sqrt{\frac{\hat{K}}{\alpha_0}}$, which is modulated by the

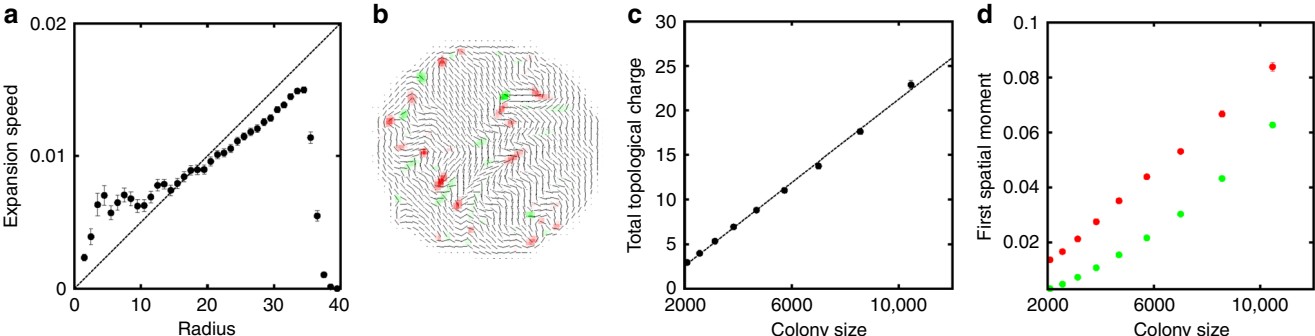

**Fig. 6** Simulations of 2D colony growth. **a** Plot of the azimuthally-averaged radial velocity within the colony, as a function of distance from the centre of the simulation box (both quantities are measured in simulation units). The dashed line indicates the 'Hubble' prediction, $v = \Lambda/2r$. The sharp drop in the curve at large radius corresponds to the change from inside the colony to outside. **b** Director field pattern and topological charge density (red, positive; green, negative) for a growing colony at a late time in the simulation. **c** Plot of the total unsigned topological charge $\int|q|d^2\mathbf{r}$ versus colony size, $\int\rho d^2\mathbf{r}$ (in simulation units). Filled circles: average over 10 simulated colonies (errors are standard error of the mean). Dashed line: linear fit. **d** Red circles: first spatial moment of positive topological charges in the colony versus colony area (in simulation units). This quantity is computed as $\int_{q>0}|r|d^2\mathbf{r}/\int_{q>0}d^2\mathbf{r}$, where $\int_{q>0}$ denotes that integration/summation only involves points with $q > 0$. Green circles: analogous quantity for negative topological charges. Parameters are: $\Lambda = 0.001$, $a_0 = 0.02$, $G = \gamma = \rho_0 = 1$, $\alpha_0 = \hat{K} = 0.01$, $\xi = 0.7$. Simulations were initialised with a radius of 10 simulation units, and the Q-tensor equal to zero with some small noise. In **a**, **c**, **d** error bars are standard errors of the mean

defect size $\sqrt{\frac{K}{a_0}}$. A more systematic study would be required to confirm this however. As $a_0$ decreases, the domain size increases until for sufficiently small $a_0$ the bending instability is no longer active and the colony resembles a nematic tactoidal droplet with tangential anchoring (Supplementary Fig. 8).

## Discussion

We have described the structural evolution of a growing 2D colony of rod-shaped E. coli until the point it buckles into the third dimension using the language of liquid crystal physics. Our system displays features that appear similar to those that are familiar from number-conserving active extensile nematics, such as tangential anchoring at the periphery, Fig. 3a, and a proliferation of $\pm\frac{1}{2}$ defects, Fig. 4. However, the underlying physical mechanisms differ. Thus, e.g., while invaginations from the boundary is a dominant mechanism of defect generation in a number-conserving active nematic embedded in an isotropic medium, we find that defects proliferated from the bulk of our colonies. The expansive growth of cells in these colonies drives new physics; we have named this system a 'Hubble' active nematic.

We have also proposed a continuum theory of such systems. Within this framework, the physics is controlled by two length scales: $l_a$ associated with the generic bending instability in dry active nematics, and $l_\sigma$ associated with the diffusive propagation of growth stress. For a colony with size $2R \ll l_\sigma$, growth stress relaxes rapidly, and the cell flow is effectively incompressible. The colony breaks up into domains of size ~$l_a$. A colony with $2R \to l_\sigma$ will accumulate stress and eventually buckle at the centre, where the accumulated stress is highest. In all cases, we expect tangential alignment at the periphery. All these predictions are in agreement with our experimental observations, and are confirmed by computer simulations of this model. More quantitatively, such simulations further reproduce our experimental finding that the total number of defects increases approximately linearly with colony area, and that positive charge defects are more peripheral than negative charge ones.

To enable even more quantitative comparison between experiments and simulations of our growing active nematic system, progress needs to be made on a number of experimental fronts. Perhaps most importantly, the nature of the friction between growing cells and their substrate, i.e. the physics underlying our parameter $\hat{\gamma}$, needs to be elucidated. There is also a need to understand the mechanics of bacterial growth into bulk agarose. It is known that the elastic properties of the surroundings have a complex effect on, e.g., the point at which a 2D colony buckles into the third dimension[17]. The detailed mechanisms for these effects are far from understood. The effect of bacterial growth rate also needs to be elucidated. Finally, it will be fascinating to develop methods for studying the growth of bacterial colonies in 3D[38], for which there are very few data at present[39]. Arriving at a predictive understanding of bacterial colony growth will constitute an advance not only in fundamental active matter physics, but also in practical fields such as food safety[40].

## Methods

**Experimental methods**. Overnight cultures of E. coli K-12 strain MG1655 were grown in M9 medium at 37 °C and shaken at 200 rpm, harvested in exponential phase, and diluted 100-fold (to an optical density of ≈0.2–0.3 at 600 nm). A number of 1.5–2 μL droplets were then inoculated onto a thin layer of 2% M9 agarose filling a polymeric frame (Thermo Scientific Gene Frame AB-0630, ≈0.25 mm thick) stuck onto a standard microscope slide, and a cover slip was sealed on top. Growing colonies were observed at 37 °C in phase contrast mode using a Nikon Eclipse E-800 microscope with a Nikon Plan Apo λ 100×/1.45 Ph3 Oil objective. Images were taken (Q-imaging Retiga 2000R camera) at regular intervals (typically 1 min). Using a published protocol[41] we obtained the time-dependent centre of mass coordinates $\mathbf{r}^i(t)$, length $L_i(t)$, and orientational unit

vector $\mathbf{v}^i(t)$ of each cell $i$. The cell diameter, $W \approx 0.9$ μm, varies little between cells or during growth. The length varies from ~2 μm just after division to ~4 μm just before. We report data from 32 colonies.

**Image recording and processing**. A schematic of the experimental set-up is shown in Supplementary Fig. 1a. For observations under phase contrast illumination intensity was minimised and exposure time varied from 20 to 100 ms. In a separate experiment, we performed confocal microscopy using fluorescent bacterial cells and agarose. For this case, the dye used for agarose staining was rhodamine B (0.02% w/v) (red channel); this was used to enhance contrast with the GFP-labelled cells (green channel) and the glass that is the black area in the images. Rhodamine B preferentially concentrates at the agarose surface. Sideways reconstruction of an image stack, Supplementary Fig. 1b, shows that cells are embedded just within the agarose beneath the glass side.

Analysis of phase contrast images was performed using ImageJ and Matlab. Initially, frames were cropped to the actual area covered by the colonies at the final stage (the field of view achievable is at least three times larger). The frames were then adjusted in brightness and contrast, and corrected for inevitable drifts (a few) in the agarose pad, especially toward the beginning of the imaging period. The latter was due to the agarose taking ~10 min to cool to room temperature, having been poured into the frame at its melting point. The shift correction was performed using ImageJ (macro StackReg[42]). Finally, the frames were segmented using the Matlab-based Schnitzcells software[41] to extract the positions, orientations and lengths of the individual cells. In order to extract velocities, the trajectories were stitched together in MATLAB using the standard Crocker and Grier code[43].

**Active field theory simulations**. Numerical simulations of Eqs. (1), (11), and (21) were performed by using finite difference methods to discretise Eqs. (1) and (21). For reasons of numerical stability, we have set the growth rate $\Lambda$ to 0 when the density was locally larger than $2\rho_0$. Note also that in Eq. (21) we need $\mathbf{v}$, rather than $\rho\mathbf{v}$ which is the natural variable in Eq. (11): to avoid problems arising from division by a very small number we have set $\mathbf{v} = 0$ when $\rho$ was below a threshold ($\rho_0/2$ worked well for our parameter set).

In wet active gels, the key control parameter to determine when active turbulence sets in is $\frac{aL_s^2}{K}$, where $a$ is the active stress parameter (in our case equal to $a_0\rho$), $L_s$ is the system size, and $K$ is the liquid crystalline elasticity[6]. Active turbulence sets in when this control parameter exceeds a critical value: therefore an infinite system always develops active turbulence. On the contrary, for dry active fluids the turbulent regime sets in when $\frac{a_0\rho}{\gamma K}$ exceeds a threshold[35]. This holds for our growing system as well, and we find the threshold is ~2, as can be shown in Supplementary Figs. 8, 9, where we plot the density of points with low orientational order. This plot also shows that the density of low order regions, which we assume should scale as $l_a^{-2}$ scales linearly with $a_0$ close to the transition to active turbulence, corresponding to a scaling of $l_a \sim (a_0 + C)^{-1/2}$ in this regime (this is consistent with the behaviour found in the ref. [35]). More data are required to firmly prove this scaling, as the range of $a_0$ we investigated is relatively narrow.

The fact that in our experiments there are microdomains within our colonies suggests we are in the regime where nematic order is disrupted and defects arise (active turbulence). Therefore a suitable regime to describe experiments within our active field theory requires $\frac{a_0\rho}{\gamma K} \geq 2$.

**Code availability**. The Matlab codes used for the analysis and in-house code written for active field theory simulations are available from the corresponding author upon request.

## Data availability

Data on bacterial colony growth are available from the corresponding author upon request. The tracked colony data used in this manuscript are also available at https://dx.doi.org/10.7488/ds/2444.

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

## Acknowledgements

We thank A. Dawson for help with microbiological lab work, and M. E. Cates, A. Doostmohammadi, M. C. Marchetti, J. Toner, B. Wacław and J. M. Yeomans for helpful discussions. The majority of the work was funded by EPSRC grant EP/J007404/1 and ERC grant AdG 340877 PHYSAPS. D.D. acknowledges funding from EPSRC and SUPA. J.S.L. acknowledges EU intra-European fellowship 623637 DyCoCoS FP7-PEOPLE-2013-IEF.

## Author contributions

W.C.K.P. and D.M. designed research. D.D.A., A.T.B. and A.F.M. performed experiments. M.L.B., F.D.C.F., J.S.L., D.M. performed simulations. All authors analysed the results and wrote the manuscript.

## Additional information

**Competing interests:** The authors declare no competing interests.

