## [Peer Review File · Nature Communications]

Reviewers' comments:

Reviewer #1 (Remarks to the Author):

In this manuscript the authors study the growth of single-layers of rod-shaped *Escherichia coli* colonies confined between soft agarose and a glass slide. They observe the growing colonies experimentally using optical and fluorescence microscopy and microscopy, and analyze this as a liquid crystal system. The authors find that growth induces flow and alignment of cells into local domains in the colony interior, though the orientation is tangential at the periphery. The authors have attempted to develop a theoretical understanding of this phenomenon, and drawn analogies of a growing bacterial colony with the expanding universe.

Despite the excitement with which I started reading the manuscript, I was disappointed by the end of the read that the term "Hubble active nematics" was used without the thought that it had deserved. The comparison, or the lack of it, between a growing bacterial colony and Hubble expansion, despite apparent similarities, could have been an interesting premise for a good paper. As much as I would have loved to see this comparison play out – which would have added a certain neat element to this increasingly explored field (physics of growing colonies) – over the length of the manuscript, I felt that the authors have used "Hubble" as a catchy namesake in the manuscript title, without a clear appreciation of what it means. The lack of discussion on why a growing bacterial colony is comparable (or NOT !) to an expanding universe made for a gaping hole in the manuscript. By using it in a loose sense and erroneously, compounded by poor phrasing, the attempt made by the authors to drive home an analogy, has done more damage than good to the manuscript. The second major weakness of the paper is its theoretical description, which, in contrast to what was claimed in the abstract, presents only a speculative, and incomplete picture. With great anticipation I was looking forward that the authors build up a case of comparing the experimental results with the theoretical modeling, which unfortunately, appear as two isolated entities in the paper. Finally, a number of experimental details, including control tests and protocols are missing in this manuscript – which should have been reported for ensuring the viability of these results, and experimental verification/ reproduction of this work in future.

Due to the serious concerns stated above (and explained in details below), I cannot recommend this manuscript for publication in Nature Communications. Below are my specific comments:

1) The authors present a review of available models of active nematics in the opening paragraphs of the manuscript. Though relevant in bits, a chunk of this is textbook literature which does not find a direct relevance to the work presented by the authors. If the authors have done this to motivate the readers about the novelty of their theoretical model/ and what it can prediction (see: "However, no theory of growing active nematics has yet been tested against experimental data. Here we provide such data by analysing growing *E. coli* colonies as 'living liquid crystals' "), the stated motivation and the results that follow are incongruous in a few ways:

a) the authors claim that there is "no theory" to support such experiments – this contradicts their initial dwelling on different theoretical premises available to address such a system. I believe that authors need to put some work in identifying the open/unaddressed problems that are relevant to the problem at hand (as a reader this was missing to me), and how their model can help solve these. This can then be related to what has already been done (some of which authors have put down in the opening paragraphs).

b) the authors claim, in the abstract and in introduction, that they provide theoretical modelling for such systems – this is a bit of stretch since what the authors have done in this manuscript is to provide certain facets/building blocks towards a potential theory to describe such a system. Interestingly (and contradicting their previous claims), at the start of the section on theoretical description, that authors state "Instead, we delineate a number of components that must feature in such a theory and point out their implications." The reviewer suggests that the authors maintain

a clear set of claims right from the outset, lest one runs chances of losing credibility later on.

c) for the sake of uniqueness, the reviewer would like to suggest that the authors find an alternative term for their system, as "living liquid crystals", has been taken and used in the context of motile bacterial cells in a liquid crystal bulk (see: <http://www.pnas.org/content/111/4/1265>).

2) The terminology "model B active matter" is misleading and out of context. The classification model A, model B etc. was introduced by Hohenberg and Halperin in the context of phase dynamics (not dynamical systems as the authors erroneously point out) to discriminate between different models of coarsening. The name "model B", in particular, indicates certain class of coarsening scenarios where the dynamics of the order parameter is governed by the Cahn-Hilliard equation. Interestingly, none of the papers that the authors cite as examples of "model B active nematics" focus on phase dynamics (which is instead the topic of some recent article by Cates' group) or uses Cahn-Hilliard equation (if nothing else as a mathematical device the handle an interface, as one of the author has done in the context of active droplets). The choice of labeling mass-conserving active systems with a name that is already used for something else appears therefore misleading and inaccurate.

3) A further, even more inaccurate is the attempt of the authors to link the results of this paper with cosmic expansion, and the bacterial growth rate with the cosmological constant. The authors should have been cautious while drawing such parallels, when, starting from the introduction, they state that "Alluding to the cosmological constant we call this class of systems 'Model Λ ' (dry) active nematics.". It is not clear why the authors actually wish to refer to, or what they know about the cosmological constant. Are they claiming that Eq. (1) maps into Friedmann's equation, with the bacterial growth rate playing the role of the cosmological constant? If that is the case, then the claim is incorrect as can be verified. If, on the other hand, the analogy should be understood only in a loose sense, without any pretension of scientific rigor, then I do not believe that this warrants a spot in the main title of the paper, not to mention that I miss a justifiable motivation for the authors make it in the first place. In its present form, the unsupported claims and comparisons, do not help improve the appeal of the manuscript and raises serious questions if the results, per se, cannot stand on own merit?

4) Colony morphology: The authors observe in their experiments that the overall morphology of the colony (elongated vs circular) is a function of time/growth stage, and conclude that initially the colonies are elongated and then they turn less anisotropic in shape. Do the authors make this as a general conclusion, which to the reviewer sounds like one? The absence of additional information, in particular, the rigidity of the substrate, aspect ratio of the growing cells, and the changes in doubling time will have non-trivial effects on the colony geometry. The authors do not report i) which dye they have used for staining the agarose layer, and ii) any control test with cells that were exposed to agarose stained with fluorescent dyes? Have the authors assessed the viability and toxicity of the bacterial cells under these conditions? Did they carry out the fluorescent confocal imaging under the same temperature conditions as in the phase imaging?

5) The authors, in the supplementary section, state that "this statistic has significant colony to colony variations" – have the authors made any attempt to quantify this variation to rule out any systematic error in their experiments? The authors need to standardize their description of "growth rate" – which in every case should have the dimension of $1/[\text{Time}]$. In Fig. 2 the authors use nm/s as a unit of growth rate – what does this refer to? Surprisingly, in SI Fig. S2(b) the units are correct, though the values are more than an order of magnitude lower. This is confusion, to say that least, and the authors need to show some diligence in representing the data consistently. Furthermore, the representation of the 32 colony data makes it difficult to study. What can the authors state about the "hump" in the growth rate curve reported in SI Fig. S2(b)? What implications will it have on their theoretical model? Finally, what do the color coding refer to in each figure? Is there a relation between the colors in (a) and (b) of Fig 3? I am a bit surprised to see that the authors have not followed any of the basics of data representation/captioning, which

makes reading the manuscript difficult.

6) The attempted connection of the results with Hubble's expansion lacks of a solid foundation. Among others, one (simple) reason that distinguishes the two settings is that bacterial micro-colonies are neither homogeneous (because the cell density is relatively higher at the middle) nor isotropic (because the expansion occurs predominantly along the radial direction). In contrast, the spatial distribution of matter in the expanding universe is homogeneous and isotopic, hence identical to all observers at a sufficiently large scale. This crucial concept in cosmology, popularly known as the "cosmological principle", is not realized in a growing bacterial colony. The Hubble parameter, that the authors erroneously put it as measure of the radial component of the velocity divided by the distance from the center of the colony, in reality corresponds to the expansion rate of the scale factor in the FRW metric. Thus the Hubble parameter describes how the distance between "any two points" in space grows in time. This referee finds frustrating having to explain the authors what they should have carefully checked and double checked before attempting such a pretentious analogy.

7) In Fig. 4 (b) – the reviewer missed any description/protocol of how the topological defect identified and characterized? Did the authors do this manually or use an automated code? Also, pertaining to data analyses, how did the authors evaluate the radial velocity of a bacteria (did the authors carry out any time averaging, if so, how?, Fig. 2 b)? How did the authors account for dividing cells at a given distance from the center of the colony?

8) Active anchoring: The reviewer does not find any justification for using the term "active" here. Why should this be "active" in any manner – are the cells locally adapting to the growth forces due to biomolecular mechanisms as response to these forces? Do the authors anticipate that this will be any different if the growth rate increases/decreases? This (again) is a flashy but extraneous term, used without much thought in the present context.

9) The authors mention performing Monte Carlo simulations based on their experimentally derived spherocylinder parameters, however, provide no relevant information on the boundary conditions (essentially, they make no comment on how realistic these simulations are)? Although the authors state the spherocylinders are polydispersed, there is no accompanying data with it.

10) In the defect dynamics section, the authors extract values of 0.05 microns/s. What is the confidence level of this value, how to the authors extract this? Do the (optical) experimental tools allow the authors to extract nanometric values?

11) "growth stress propagates by diffusion" – The reviewer is confused what the authors mean by this, since there is no supporting explanation or demonstration of what the authors have claimed. Can the authors, for a moment, forget the jargons and explain in plain words what does it mean? Does it mean that the stress tensor has a diffusive dynamics? Or is it that Brownian diffusion facilitates stress propagation? Any evidence in the paper what actually the case is?

12) The section "Theoretical description" reads extremely vague and speculative, compared to what the authors claim in their abstract. It is highly unclear where do the reported length scales come from? Are they general to active nematics or do they represent a special case to bacterial colonies? Why, according to the authors, does the active stress depend only on the elastic modulus of the cell envelope and not, for instance, on the growth rate? How do the authors account for the extra cellular matrix – will this not contribute to the elasticity of the cell modulus? How do the authors account for this? Importantly, the referee completely misses any mapping in this regard, between the theoretical description and the experiments that they report.

Reviewer #2 (Remarks to the Author):

The paper analyzes the growth of bacterial colonies in a quasi two-dimensional geometry. Experimental results report the linear growth of the colony up to certain radius, before a buckling instability directs the growth towards the third dimension. In addition, several measures are provided relative to the defect dynamics and their accumulated distinctive populations depending on their topological charges (positive vs. negative).

From a theoretical point of view, rather than a detailed model, authors discuss several aspects that they claim should be included in a full theory.

The paper is well-written and conveniently put in the context of the research in the field of active nematics. This line of research is presently very appealing beyond the common reference to cytoskeletal- based preparations. In this regard, it's worth remembering recent reports published in Nature and Nature Physics from Silberzan, Ladoux and Sano's groups. The paper by Dell'Arciprete seems to follow similar lines but referring to bacterial colonies. Authors have built a sustained experience in the field.

I think the paper should certainly be published since, as far as I know, observations are reasonably original, but I'm not sure this work represents a major advance in the field, in the sense of stimulating people to look differently at the problem of bacterial colonies growth. Probably a full account of theory and a deeper comparison with experimental observations would help to better appraise the novelties of the approach proposed.

Major comments:

1) I was somehow surprised to see that any kind of fingering instability is apparently suppressed in the reported experiments. Is that a question of the kind of bacteria used, the way the experiments are conducted, or the particularly analyzed growth conditions?

2) The role of defects is not considered at all as a way to relax active stress. It is known that this is the common rationale, at least in wet active nematics. Is that different here once the authors claim their system to be dry? A discussion on this issue would be appreciated.

3) Apparently viscous effects are completely neglected, but as authors recognize themselves, it seems that wet and dry systems are only extreme cases, and probably both sort of effects concomitantly come to play a role in the dynamics of active systems. Why are not included here?

Minor comments:

1) Please check caption of Fig. 5: I believe reference to scheme (b) in last sentence is wrong.

2) In subsection "Two length scales" there is a scaling estimation of the stress propagation length, and further authors say that such a estimated value is similar to experiments. I have not been able to find in the experimental part of the paper any reference to this length (50 microns).

Reviewer #3 (Remarks to the Author):

Arciprete et al. consider two-dimensional E. Coli colonies on a substrate. They describe the shape dynamics of the colonies. Rod-shaped colonies initially increase their aspect ratio as they grow and then converge to an approximately spherical shape. They characterize growth speed and the expansion of the colony as a function of the colony size. An essential point of the authors is that the expansion speed of the colony increases linearly with colony size, which is reminiscent of the expanding universe. They characterize nematic alignment and track the topological defects of the expanding colonies: $\pm 1/2$ defects are produced at constant rate and $+1/2$ defects are "propelled" toward the boundary. They suggest that the observed defect dynamics is reminiscent of extensile active nematics. Their main claim is, however, that the growing bacterial colony belongs to a different universality class than "active nematics", which they refer to "Hubble active nematics" due the similarity to the expanding universe.

General comments:

The work is of good quality, clearly structured and most parts are understandable. The question of

a new universality class is novel and might be of interest for a broad interdisciplinary readership. Within the specific field of active matter, this work might represent a seminal contribution. However, I think that there are several points, which require revisions in terms of clarity. My main criticism refers to the key finding of a novel universality class. My recommendation will predominantly depend on how the authors reply to this issue.

Major comments:

Make clear(er) how Fig. 2(b) and Fig. 2(c) are connected. These figures are key in the paper, so the reader should immediately get it without spending too much time.

More importantly, please comment on the experimental error of the dots in Fig. 2(c). Convince the reader that this is not a cloud of points where growth rates fluctuate by \sim a factor of two. The linear "fit" is not very convincing to me and to my understanding it should support one of the key findings in the manuscript. The authors have to strengthen this point by either obtaining data points at lower/higher growth rates, adding error bars, both, etc ...

Fit figure 4(b):

Do you include all data points into the fitting procedure, also the one at small number of bacteria? Since one would expect a power law in the asymptotics of large numbers of bacteria, could you check what the coefficient is in the range 100 to 1000? It looks to me that it perfectly scales with N^1 . Maybe not, but please check. What is the theoretical prediction for active nematics?

Minor comments:

Introduction:

What does "extremes of a continuum" mean?

I am not sure whether the authors use the definition of Model B correctly in their introduction. To my knowledge, Model B has no sink/source terms that break conservation of mass. Especially in the case of cell colonies (which do not care about thermal physics and chemical potentials), e.g. phase separated or ordered states can easily get destabilized despite of reaction equilibrium (homeostasis).

Paragraph right before section 'Hubble nematics':

Growth by cell divisions includes two time scales: the time between two successive division events and the actual time of division. It seems that Fig. 2(a) depicts the later or both as you write "measurement of cell length as a function of time".

Have you tracked each cell continuously in time without losing the tracking of cells? Then your measure includes both time scales. But please comment.

Begin of section 'Hubble nematics':

I would recommend to explicitly say that you write down equations based on/consistent with experimental observations.

Before Eq. (1) say explicitly that you focus on the exponential growing regime by writing a gain term $\propto \lambda * \rho$.

Say where the factor of 2 comes from in the radial speed v_r .

For the time regime investigated (inset Fig. 1a) colonies appear to become never spherical. Please

make clear what the algorithms extracts as you are plotting along the "radius" in Fig. 2(a,b).
Mention this in the caption or the main text.

Moreover, please find a clear (say less confusing) terminology for the "rates", broadly speaking. I would recommend using the word "speed" if the quantity has units of [nm/s] and "rate" if the unit is [1/s]. At least, please introduce different names for different quantities, (e.g. not growth rate in Fig. 2 (a) and Fig. 2(c)).

Eq. (3): Say in words why you need the function f^i that smears out the individual bacteria. Explain why you can't you simplify use some "center-of-mass coordinate" etc... ?

We are very grateful to the Reviewers for the feedback received, which we believe have led to an improvement of our work.

Following is a point-by-point reply to the Reviewers' comments. We hope that our revised manuscript can now be published in Nature Communications.

Please note that the main changes (other than minor editing ones) in the revision are highlighted in red, in both the main text and Supplementary Information.

Reply to Reviewer #2

OVERALL COMMENT:

"The paper analyzes the growth of bacterial colonies in a quasi two-dimensional geometry. Experimental results report the linear growth of the colony up to certain radius, before a buckling instability directs the growth towards the third dimension. In addition, several measures are provided relative to the defect dynamics and their accumulated distinctive populations depending on their topological charges (positive vs. negative).

From a theoretical point of view, rather than a detailed model, authors discuss several aspects that they claim should be included in a full theory.

The paper is well-written and conveniently put in the context of the research in the field of active nematics. This line of research is presently very appealing beyond the common reference to cytoskeletal-based preparations. In this regard, it's worth remembering recent reports published in Nature and Nature Physics from Silberzan, Ladoux and Sano's groups. The paper by Dell'Arciprete seems to follow similar lines but referring to bacterial colonies. Authors have built a sustained experience in the field.

I think the paper should certainly be published since, as far as I know, observations are reasonably original, but I'm not sure this work represents a major advance in the field, in the sense of stimulating people to look differently at the problem of bacterial colonies growth. Probably a full account of theory and a deeper comparison with experimental observations would help to better appraise the novelties of the approach proposed."

RESPONSE:

We thank the reviewer for considering our work of interest and for judging that it should certainly be published. The reason why we feel it is a major advance in the field is that it provides the first systematic set of quantitative experimental results on a growing bacterial colony. On the other hand, we have fully taken the referee's suggestion on board, and now provide simulations of a full continuum theory built on the elements discussed in our original submissions. The results are in good agreement both with the previous, more qualitative, discussion, and with our experiments. The comparison with experiment is now deeper, as we can describe defect proliferation and topological charge segregation with our numerical simulations (Fig. 6 in the amended version). We feel that this has improved our manuscript and are grateful for this suggestion.

COMMENT:

"Major comments:

1) I was somehow surprised to see that any kind of fingering instability is apparently suppressed in the reported experiments. Is that a question of the kind of bacteria used, the way the experiments are conducted, or the particularly analyzed growth conditions?"

RESPONSE:

Fingering instabilities do not arise here because the growth rate is approximately constant in space in our experiments (see Fig. 2a and corresponding discussion). For instance, the theory work in F. D. C. Farrell, O. Hallatschek, D. Marenduzzo, and B. Waclaw, Phys. Rev. Lett. 111, 168101 (2013) shows that such instabilities arise only when the nutrient field and its impact on growth rate is considered.

More generally, a fingering instability may also arise if the growth rate depends on density, see e.g. the work on undulation instabilities in tissues M. Basan, J.-F. Joanny, J. Prost, and T. Risler, Phys. Rev. Lett. 106, 158101 (2011).

We have now clarified this point and added a reference to the latter work.

COMMENT:

"2) The role of defects is not considered at all as a way to relax active stress. It is known that this is the common rationale, at least in wet active nematics. Is that different here once the authors claim their system to be dry? A discussion on this issue would be appreciated."

RESPONSE:

We note we actually did mention this issue, although admittedly very briefly, saying that defects may have a role in relaxing growth stresses (final sentence in the paragraph after Eq. 12). This was motivated by the observation that the lengthscale obtained by multiplying the velocity of the $+1/2$ defects by the growthrate is, quite intriguingly, close to the lengthscale at which buckling occurs, i.e. 50 microns.

The reason why we only mentioned this in passing is that in our dry case any compressibility is known to be the main avenue to relax stresses, at least in computer simulations [see, e.g., F. D. C. Farrell, O. Hallatschek, D. Marenduzzo, and B. Waclaw, Phys. Rev. Lett. 111, 168101 (2013)]. If the system is purely incompressible, then defects are a viable alternative for stress relaxation. We also note that the two modes may be related as the point at which the colony buckles into the third dimension (where arguably stresses are highest) approximately coincides with the location of defects (mainly $-1/2$ defects in our case). This plot was included in the original submission, but we have now commented it in more detail.

COMMENT:

"3) Apparently viscous effects are completely neglected, but as authors recognize themselves, it seems that wet and dry systems are only extreme cases, and probably both sort of effects concomitantly come to play a role in the dynamics of active systems. Why are not included here?"

RESPONSE:

We agree with the reviewer that wet and dry systems are only extreme cases, and now have clarified that they are limiting cases of a continuum of possible theories (page 1, column 1 of the revised version).

As the referee notes we have taken the dry limit, which is physically motivated by our experimental setup where bacteria are partially embedded in the agar and very close to the glass coverslip (see Fig. S1). Including a viscous contribution, where the viscosity is η , would lead to an additional "screening length scale", $\sqrt{\eta/\hat{\gamma}}$ in our notation. This is the length over which long range hydrodynamic interactions are screened, and our dry description will hold. In our case due to the strong interaction with the glass slide the screening length is of the order of the bacterial cell size, justifying our dry limit.

COMMENT:

"Minor comments:

1) Please check caption of Fig. 5: I believe reference to scheme (b) in last sentence is wrong."

RESPONSE:

Thanks for flagging this up. We have now corrected this sentence in the caption.

COMMENT:

"2) In subsection "Two length scales" there is an scaling estimation of the stress propagagtion length, and further authors say that such a estimated value is similar to experiments. I have not been able to find in the experimental part of the paper any reference to this length (50 microns)."

RESPONSE:

This is the length scale over which the colony buckles into the third dimension -- we can estimate this via microscopy, and have now mentioned this in the section describing colony morphology.

Reply to Reviewer #3

OVERALL COMMENT:

"The work is of good quality, clearly structured and most parts are understandable. The question of a new universality class is novel and might be of interest for a broad interdisciplinary readership. Within the specific field of active matter, this works might represent a seminal contribution.

However, I think that there are several points, which require revisions in terms of clarity. My main critizm refers to the key finding of a novel universality class. My recommendation will dominantly depend how the authors reply to this issue."

RESPONSE:

We thank the reviewer for judging our work novel and potentially a seminal contribution to active matter. We note that this is indeed to our knowledge the first work with extensive quantitative experimental data on the liquid crystalline order of growing bacterial colonies.

We are also grateful to the reviewer for her/his constructive criticism. We have now taken all her/his comments into account, as detailed below in our point-by-point reply.

COMMENT:

"Major comments: Make clear(er) how Fig. 2(b) and Fig. 2(c) are connected. These figure are key in the paper, so the reader should immediately get it without spending too much time."

RESPONSE:

Fig. 2(b) shows the velocity versus position for all our data (i.e., average over all colonies) -- this is the most important plot in Figure 2, as it is the analogue of Hubble's law for bacterial colonies. Fig. 2(c) measures the spread over the population (i.e., it quantifies intrinsic noise): each point gives the value of H measured for a **single** colony. We have now expanded the discussion in the text to make this clearer.

COMMENT:

"More importantly, please comment on the experimental error of the dots in Fig. 2(c). Convince the reader that this is not a cloud of points where growth rates fluctuates by ~ a factor of two. The linear "fit" is not very convincing to me and to my understanding it should support one of the key findings in the manuscript. The authors have to strengthen this point by either obtaining data points at lower/higher growth rates, adding error bars, both, etc"

RESPONSE:

Error bars in the points in Fig. 2(c) are not easy to include because, as now clarified, each point comes from the fit of a single colony. In other words, these points already give a measure of the intrinsic noise in our colonies. For the same reason, the growth rate cannot be changed in a controlled way, as the range we can sample is the one which naturally occurs in our colony population.

At the same time, we should highlight that the really key point for the paper in Fig. 2 is that the velocity and positions are linearly related (see Fig. 2(b), which considers the average over all our colonies). Fig. 2(c) explains much of the variability found in the single colony fits (a 2-fold change in H) as due to the different growth rate. The key point of Fig. 2(c) is to show that there is a correlation between the value of H in an individual colony and the value of Λ in that same colony, as both can vary about 2-fold.

To address the referee's question to the best of our abilities, we have now rendered our statement more quantitative, and calculated the p-value for the correlation found in the fitting. We found a Pearson correlation of $r=0.7726$ and a p-value $p<0.00001$ for the dataset in Fig. 2(c), which correspond to a highly statistically significant correlation. As an additional calculation for the reviewer, we have also provided a simple-minded estimate of error bars in Fig. 2(c) (see ExtraFig1.png). These are estimated via 6 snapshots of each colony, at different times in the evolution (and assuming measurements to be uncorrelated for simplicity). By using these error bars (standard deviation of the mean) we have generated a distribution of Pearson correlations via a Monte Carlo procedure: the median, first quartile and third quartile are respectively $r=0.665$, $r=0.585$, and $r=0.725$ (suggesting that error bars may have been slightly overestimated). The corresponding p-values are respectively $2e-05$, $3e-04$ and <0.00001 - in all cases statistically significant. We do not include this more complicated analysis in the main text, as the error bars are likely overestimated - the simpler analysis now included in the text is sufficient to make the point that data are linearly correlated.

The first-principle simple prediction that $H=0.5\Lambda$ is also quantitatively close to the data lending further validation to the statement that the single colony data in Fig. 2(c) are highly correlated.

COMMENT:

"Fit figure 4(b):

Do you include all data points into the fitting procedure, also the one at small number of bacteria? Since one would expect a power law in the asymptotics of large numbers of bacteria, could you check what the coeff. is in the range 100 to 1000? It looks to me that it perfectly scales with N^1 . Maybe not, but please check. What is the theoretical prediction for active nematics?"

RESPONSE:

We already discard data for very small number of bacteria (these do not affect much the result though). A fit in the range 100 to 1000 bacteria yields indistinguishable results from the one presented (exponent 0.93). We agree that the fit is fully compatible with N^1 , and we have now said it more clearly (the quoted exponent is the result of a linear fit in the log-log plane, and is very close to 1). The theory of active nematics is consistent with a linear scaling, in the regime of "active turbulence", as the distance between defects (hence their density) is fixed by the active length scale l_a . We have now made this clearer, and the linear behaviour is confirmed by our new numerical simulations (see Fig. 6c).

COMMENT:

"Minor comments:

Introduction:

What does "extremes of a continuum" mean?"

RESPONSE:

Here we mean that the dry and wet theory are to be considered as limiting cases ("extremes"). Introducing both friction and viscosity would give a family of interpolating theories (the "continuum"). We have now rephrased to clarify.

COMMENT:

"I am not sure whether the authors use the definition of Model B correctly in their introduction. To my knowledge, Model B has no sink/source terms that breaks conservation of mass.

Epecially in the case of cell colonies (which do not care about thermal physics and chemical potentials), e.g. phase separated or ordered states can easily get destabilized despite of reaction equilibrium (homeostasis)."

RESPONSE:

We agree that strict model B has no sink/source, so that it applies literally to the case of previous theory and experiments where particle numbers is conserved exactly, which is the vast majority of the cases studied in the literature thus far. We mentioned as a distinct class the case with homeostasis where the number of particles is only constant on average - in our current case, there is no reaction equilibrium (the particle number keeps increasing) so that the problem is still qualitatively distinct. To avoid inaccuracies, and to address the reviewer's comments, instead of using Model B/Model Lambda we now refer to the older model as 'number-conserving' and to ours as 'non-number-conserving'.

COMMENT:

*"Paragraph right before section 'Hubble nematics':
Growth by cell divisions includes two time scales: the time between two successive division events and the actual time of division.
It seems that Fig. 2(a) depicts the later or both as you write "measurement of cell length as a function of time". Have you tracked each cell continuously in time without losing the tracking of cells? Then your measure includes both time scales. But please comment."*

RESPONSE:

We tracked individual cells and their length, $L(t)$, so that we know their area as a function of time (see SI, Section S-II-B). As described in the SI, the growth rate is measured as $1/A \, dA/dt$, Eq. (S1), so the relative increase of area, or bacterial mass. With this definition, there is no need to specify whether the cell is growing or dividing at any time. Note that we have now corrected the units in Fig. 2(a), so together with the specification in the SI we trust our procedure and calculations are now clear.

COMMENT:

*"Begin of section 'Hubble nematics':
I would recommend to explicitly say that you write down equations based on/consistent with experimental observations. Before Eq. (1) say explicitly that you focus on the exponential growing regime by writing a gain term $\propto \lambda * \rho$.
Say where the factor of 2 comes from in the radial speed v_r ."*

RESPONSE:

Agreed - now added equations are based on experimental observations. However Eq. (1) does not necessarily correspond to the exponential regime (as λ may not be constant). The factor of 2 in the radial speed comes from working in radial coordinates in 2D (now mentioned).

COMMENT:

*"For the time regime investigated (inset Fig. 1a) colonies appear to become never spherical. Please make clear what the algorithms extracts as you are plotting along the "radius" in Fig. 2(a,b). Mention this in the caption or the main text.
Moreover, please find a clear (say less confusing) terminology for the "rates", broadly speaking. I would recommend using the word "speed" if the quantity has units of [nm/s] and "rate" if the unit is [1/s]. At least, please introduce different names for different quantities, (e.g. not growth rate in Fig. 2 (a) and Fig. 2(c))."*

RESPONSE:

The radius is the distance from the centroid of the colony (clarified in caption of Fig. 2) - now we have included in the SI a detailed protocol for the data analysis of Fig. 2.

Also many thanks for spotting this glitch in our notation. We now use "speed" and "rate" as suggested by the reviewer.

COMMENT:

"Eq. (3): Say in words why you need the function f^i that smears out the individual bacteria. Explain why you can't you simplify use some "center-of-mass coordinate" etc ?"

RESPONSE:

The function f^i takes into account the rod-like nature of the bacteria, and is necessary to find the coarse grained density, and, more importantly, Q tensor field. This would be in general less accurate if we were using other coarse graining procedure which disregard the bacterial geometry, for instance via circular Gaussians with centre in the centre-of-mass of each bacterium. We have now clarified this (footnote [28]).

Reply to Reviewer #1

OVERALL COMMENT:

"Despite the excitement with which I started reading the manuscript, I was disappointed by the end of the read that the term "Hubble active nematics" was used without the thought that it had deserved. The comparison, or the lack of it, between a growing bacterial colony and Hubble expansion, despite apparent similarities, could have been an interesting premise for a good paper.

As much as I would have loved to see this comparison play out -- which would have added a certain neat element to this increasingly explored field (physics of growing colonies) -- over the length of the manuscript, I felt that the authors have used "Hubble" as a catchy namesake in the manuscript title, without a clear appreciation of what it means. The lack of discussion on why a growing bacterial colony is comparable (or NOT !) to an expanding universe made for a gaping hole in the manuscript. By using it in a loose sense and erroneously, compounded by poor phrasing, the attempt made by the authors to drive home an analogy, has done more damage than good to the manuscript. The second major weakness of the paper is its theoretical description, which, in contrast to what was claimed in the abstract, presents only a speculative, and incomplete picture. With great anticipation I was looking forward that the authors build up a case of comparing the experimental results with the theoretical modeling, which unfortunately, appear as two isolated entities in the paper. Finally, a number of experimental details, including control tests and protocols are missing in this manuscript -- which should have been reported for ensuring the viability of these results, and experimental verification/ reproduction of this work in future.

Due to the serious concerns stated above (and explained in details below), I cannot recommend this manuscript for publication in Nature Communications."

RESPONSE:

We feel that the referee's dismissal of our work may be more based on style than on scientific issues. In particular, the reviewer seems to base her/his rejection mainly on a refusal to accept our choice of terminology and of the way in which we used Hubble's law as an analogy for our bacterial colonies. We agree that one cannot expect that the details of the physics underlying the expanding universe and our growing colonies will be literally the same, and we do not believe that we have implied this in our manuscript. What we feel is intriguing is that the phenomenology has some undeniable similarities (compare Fig. 2(b) with Hubble's original plot of the galaxy velocity versus distance). As detailed below, we have now modified our title and clarified our choice of terminology so as to leave no possible remaining room for misunderstanding, and believe that this is an appropriate response to all of this referee's comments regarding the analogy to Hubble's law.

Regarding the second weakness perceived by the reviewer, we note that the main novelty of our work is the provision of new experimental data on growing bacterial colonies, and that theory and experiments were linked as our schematic theory explained the main experimental observations qualitatively. This comparison was confined to the Summary and Conclusions for conciseness. In the revised version, we have now performed simulations of a full model based on the elements we originally outlined, which is in full agreement with our previous discussion, and also correctly captures the physics of defects (with segregation between positive and negative defects). This has also now made the comparison between experiments and theory more quantitative.

Finally, we have now made sure to include all experimental details in the revised SI, and thank the reviewer for flagging up some which we omitted. This would facilitate verification/reproduction of our work by others in the future, which we agree with the reviewer is important.

We therefore feel that this rebuttal and revision constructively address the critique raised by the reviewer, and hope that she/he will agree that the revised manuscript has improved and can now be accepted for publication.

COMMENTS:

"Below are my specific comments:

1) The authors present a review of available models of active nematics in the opening paragraphs of the manuscript. Though relevant in bits, a chunk of this is textbook literature which does not find a direct relevance to the work presented by the authors. If the authors have done this to motivate the readers about the novelty of their theoretical model/ and what it can predict (see: "However, no theory of growing active nematics has yet been tested against experimental data. Here we provide such data by analysing growing E. coli colonies as 'living liquid crystals' "), the stated motivation and the results that follow are incongruous in a few ways:

a) the authors claim that there is "no theory" to support such experiments - this contradicts their initial dwelling on different theoretical premises available to address such a system. I believe that authors need to put some work in identifying the open/unaddressed problems that are relevant to the problem at hand (as a reader this was missing to me), and how their model can help solve these. This can then be related to what has already been done (some of which authors have put down in the opening paragraphs).

b) the authors claim, in the abstract and in introduction, that they provide theoretical modelling for such systems - this is a bit of stretch since what the authors have done in this manuscript is to provide certain facets/building blocks towards a potential theory to describe such a system.

Interestingly (and contradicting their previous claims), at the start of the section on theoretical description, that authors state "Instead, we delineate a number of components that must feature in such a theory and point out their implications." The reviewer suggests that the authors maintain a clear set of claims right from the outset, lest one runs chances of losing credibility later on.

c) for the sake of uniqueness, the reviewer would like to suggest that the authors find an alternative term for their system, as "living liquid crystals", has been taken and used in the context of motile bacterial cells in a liquid crystal bulk (see: <http://www.pnas.org/content/111/4/1265>).

RESPONSE:

Regarding point a), we claimed that "no theory of growing active nematics has yet been tested against experimental data". We did not say that there is "no theory" for this system, which we agree would indeed be an unsubstantiated claim. We have now added "quantitative" before "experimental data" to further qualify our statement. The provision of such data which quantify orientational order within our colonies at different levels (globally, locally in the bulk or at the boundary, and in relation to topological defects) is the main new result, which was previously open/unaddressed.

Regarding point b), we take this, and we have now substantially enlarged our theory. First, we have spelled out that the experimental observations we make are indeed explained by our schematic theory. Second, we have now performed numerical simulations for a full theory based on our schematic model (see new Section "Active field theory simulations", and Figs. 6, S8 and S9). The conclusions confirm our previous, more qualitative, discussion, and render the comparison more quantitative. We feel that this addition has strengthened the paper.

Regarding terminology, we use now the term 'living anisotropic fluids', following nomenclature in Ref. [3] ('living fluids').

COMMENT:

"2) The terminology "model B active matter" is misleading and out of context. The classification model A, model B etc. was introduced by Hohenberg and Halperin in the context of phase dynamics (not dynamical systems as the authors erroneously point out) to discriminate between different models of coarsening. The name "model B", in particular, indicates certain class of coarsening scenarios where the dynamics of the order parameter is governed by the Cahn-Hilliard equation. Interestingly, none of the papers that the authors cite as examples of "model B active nematics" focus on phase dynamics (which is instead the topic of some recent article by Cates' group) or uses Cahn-Hilliard equation (if nothing else as a mathematical device the handle an interface, as one of the author has done in the context of active droplets). The choice of labeling mass-conserving active systems with a name that is already used for something else appears therefore misleading and inaccurate."

RESPONSE:

Our choice of terminology was to use model B in a broader sense to refer to "number-conserving" systems, as more compact. We acknowledge that it may be potentially misleading hence we have now reverted to "number-conserving" to avoid any possible misunderstanding and inappropriate comparison with coarsening theories (which indeed have no relevance here). We note that Reviewer #3 also mentioned the possible misleading nature of model B terminology, but judging this point to be minor.

COMMENT:

"3) A further, even more inaccurate is the attempt of the authors to link the results of this paper with cosmic expansion, and the bacterial growth rate with the cosmological constant. The authors should have been cautious while drawing such parallels, when, starting from the introduction, they state that "Alluding to the cosmological constant we call this class of systems 'Model A' (dry) active nematics.". It is not clear why the authors actually wish to refer to, or what they know about the cosmological constant. Are they claiming that Eq. (1) maps into Friedmann's equation, with the bacterial growth rate playing the role of the cosmological constant? If that is the case, then the claim is incorrect as can be verified. If, on the other hand, the analogy should be understood only in a loose sense, without any pretension of scientific rigor, then I do not believe that this warrants a spot in the main title of the paper, not to mention that I miss a justifiable motivation for the authors make it in the first place. In its present form, the unsupported claims and comparisons, do not help improve the appeal of the manuscript and raises serious questions if the results, per se, cannot stand on own merit?"

RESPONSE:

We certainly do not wish to suggest that Eq. (1) directly maps onto Friedman's equation. What we mean is that Fig. 2(b) is strikingly similar to Hubble's original fit of galaxy velocity versus mutual distance. We have made this point even more clearly now in the introduction, and modified the sentence highlighted by the reviewer, including a footnote. To take into account the referee's feedback we have also removed 'Hubble' from the title, retaining the discussion of this broad analogy (with clarifications) in the text.

COMMENT:

"4) Colony morphology: The authors observe in their experiments that the overall morphology of the colony (elongated vs circular) is a function of time/growth stage, and conclude that initially the colonies are elongated and then they turn less anisotropic in shape. Do the authors make this as a general conclusion, which to the reviewer sounds like one? The absence of additional information, in particular, the rigidity of the substrate, aspect ratio of the growing cells, and the changes in doubling time will have non-trivial effects on the colony geometry. The authors do not report i) which dye they have used for staining the agarose layer, and ii) any control test with cells that were exposed to agarose stained with fluorescent dyes? Have the authors assessed the viability and toxicity of the bacterial cells under these conditions? Did they carry out the fluorescent confocal imaging under the same temperature conditions as in the phase imaging?"

RESPONSE:

The conclusion holds at least for the bacterial strains we have used (a wild-type MG1655, and a GFP-labeled version of the same) and for agarose gels of 2% volume fraction. Inspection of the results in Ref. [17] suggests that qualitatively similar results should hold at least within the range of 1%-2.5% agarose. The average aspect ratio of cells just after division is 2 (cell width ~ 1 micron, cell length ~ 2 microns). We have now gathered all these pieces of information in the first paragraph in the Experimental Observation section (before they were given at different places in the manuscript). The doubling time depends on time, the way in which it does so can be seen in Fig. S2.

For the confocal imaging, the agar the bacteria is in contact with has been dyed with rhodamine B (0.02% w/v) (red channel) for enhancing contrast with the cells (green channel) and the glass that is the black area in the two side projections. We have now mentioned this in the SI. No dye was used in the phase contrast experiments which we used to analyse growth morphology in the paper. Additionally, this dye is standard and known to not interfere with cell viability (as no toxic effects were ever reported in the literature). Temperature conditions were the same in confocal and phase contrast imaging.

We have now clarified that although concordance with the theory and selected experiments in the lab with different conditions suggest this dynamical pathway to be generic, we can only state this for sure for the range of conditions we have used.

COMMENT:

"5) The authors, in the supplementary section, state that "this statistic has significant colony to colony variations" - have the authors made any attempt to quantify this variation to rule out any systematic error in their experiments? The authors need to standardize their description of "growth rate" - which in every case should have the dimension of 1/[Time]. In Fig. 2 a the authors use nm/s as a unit of growth rate - what does this refer to? Surprisingly, in SI Fig. S2(b) the units are correct, though the values are more than an order of magnitude lower. This is confusion, to say that least, and the authors need to show some diligence in representing the data consistently. Furthermore, the representation of the 32 colony data makes it difficult to study.

What can the authors state about the "hump" in the growth rate curve reported in SI Fig. S2(b)? What implications will it have on their theoretical model? Finally, what do the color coding refer to in each figure? Is there a relation between the colors in (a) and (b) of Fig 3? I am a bit surprised to see that the authors have not followed any of the basics of data representation/captioning, which makes reading the manuscript difficult."

RESPONSE:

We are unsure how to analyse the variation to detect systematic errors a posteriori. Systematic errors were minimised by controlling the system conditions and parameters.

Fig. 2(c) does show that the variation in growth rate (which is likely due to intrinsic noise) is linked to variation in the bacterial velocity as predicted by the simple law that $H \sim 0.5 \lambda$, which suggests that colony-to-colony variation are, at the very least, internally consistent.

The 'hump' in Fig. S2 shows changes over *time* - the theoretical model can deal with this (as with changes over space, at least in principle). Instead, the relation $H = 0.5 \lambda$ assumes constant growth rate over space, which is approximately the case (Fig.~2(a)): changes over time are not relevant for its validity.

Different colours in Figs.~S2 and Fig.~3 refer to different colonies, as was mentioned in the caption to Fig. S2. Besides this, we have specified that there is no relation between the colours used (i.e., we did not attempt to match colonies in the two figures, as this would be almost impossible to discern).

We have also corrected the notation and now refer to growth speed and rate. We trust that the current presentation makes this all clear.

COMMENT:

"6) The attempted connection of the results with Hubble's expansion lacks of a solid foundation. Among others, one (simple) reason that distinguishes the two settings is that bacterial microcolonies are neither homogeneous (because the cell density is relatively higher at the middle) nor isotropic (because the expansion occurs predominantly along the radial direction). In contrast, the spatial distribution of matter in the expanding universe is homogeneous and isotropic, hence identical to all observers at a sufficiently large scale. This crucial concept in cosmology, popularly known as the "cosmological principle", is not realized in a growing bacterial colony.

The Hubble parameter, that the authors erroneously put it as measure of the radial component of the velocity divided by the distance from the center of the colony, in reality corresponds to the expansion rate of the scale factor in the FRW metric. Thus the Hubble parameter describes how the distance between "any two points" in space grows in time. This referee finds frustrating having to explain the authors what they should have carefully checked and double checked before attempting such a pretentious analogy."

RESPONSE:

We reiterate that we did not mean that the physics behind the expanding universe and our growing colonies was the same. We trust that with the specification in the revised Introduction this is now clear.

At the same time, we note that this comment in 6. is not fully accurate. In the simplest case of an incompressible colony with circular symmetry, the growth and radial velocity profile would correspond to an affine deformation of the colony so that the distance between any two points in space (which are advected by the flow) would indeed grow in time with the same growth law (as in the Hubble analogy).

COMMENT:

"7) In Fig. 4 (b) - the reviewer missed any description/protocol of how the topological defect identified and characterized ? Did the authors do this manually or use an automated code ? Also, pertaining to data analyses, how did the authors evaluate the radial velocity of a bacteria (did the authors carry out any time averaging, if so, how ?, Fig. 2 b) ? How did the authors account for dividing cells at a given distance from the center of the colony ?"

RESPONSE:

Regarding Fig. 4b, we used an automated code to coarse grain the bacteria, based on a segmentation procedure of optical data. This is described in the SI. More specifically, we used Eq. (10) after coarse graining and worked on this topological charge density to identify, characterise and track defects (again, with an automated code).

Regarding the data analysis in Fig. 2b, there was no time averaging, and there was no need to do anything special to account for dividing bacteria. When a bacterium divides, the parent trajectory ends and two new trajectories begin. We agree that this should be spelled out, and we have now given in the SI the detailed protocol of our data analysis in Fig. 2b, which included the following steps:

- 1) Calculate the centroid of the colony based on the unweighted bacterial coordinates, using only trajectories which remain for at least 6 frames.
- 2) Include only those bacteria within the largest circle centred on this centroid that can be inscribed in the colony.
- 3) Calculate the displacement between each consecutive frame of each of these bacteria along the radial direction. This gives the radial velocity.
- 4) Average the radial velocity over the final 10 minutes of each video prior to buckling, binning for distance from the centroid.

COMMENT:

"8) Active anchoring: The reviewer does not find any justification for using the term "active" here. Why should this be "active" in any manner - are the cells locally adapting to the growth forces due to biomolecular mechanisms as response to these forces ? Do the authors anticipate that this will be any different if the growth rate increases/decreases? This (again) is a flashy but extraneous term, used without much thought in the present context."

RESPONSE:

Active anchoring is the terminology used for this phenomenon in the case of a wet active system, where activity leads to shear flow rather than expansion flow, see e.g. the lecture series <https://arxiv.org/abs/1603.00194>. In number-conserving systems, activity comes from dipolar forces exerted by the active particles, usually in order to swim, here it comes from pushing forces due to cell elongation/division. Therefore we feel the terminology is appropriate here.

COMMENT:

"9) The authors mention performing Monte Carlo simulations based on their experimentally derived spherocylinder parameters, however, provide no relevant information on the boundary conditions (essentially, they make no comment on how realistic these simulations are) ? Although the authors state the spherocylinders are polydispersed, there is no accompanying data with it."

RESPONSE:

As stated in the SI, section S-II.F, we took cell lengths, positions and orientation from experimental snapshots, so the polydispersity is equal to that measured from experiments. For boundaries, we enclosed spherocylinders in a circle. We have clarified that the circle is fixed at all times to avoid ambiguity.

COMMENT:

"10) In the defect dynamics section, the authors extract values of 0.05 microns/s. What is the confidence level of this value, how do the authors extract this? Do the (optical) experimental tools allow the authors to extract nanometric values ?"

RESPONSE:

We track defects by coarse graining the colonies and using Eq. (10) - we feel that this is an important technical advancement, as without it, it would be indeed extremely difficult to use the optical experimental data directly to infer defect speed. There is no issue in principle with extracting 'nanometric' values, since the time difference between two frames is 1 minute, so defects move of order 3 microns/frame.

We have added though that this value should be seen as an order-of-magnitude estimate as we could only track a few defects.

COMMENT:

"11) "growth stress propagates by diffusion" - The reviewer is confused what the authors mean by this, since there is no supporting explanation or demonstration of what the authors have claimed. Can the authors, for a moment, forget the jargons and explain in plain words what does it mean? Does it mean that the stress tensor has a diffusive dynamics? Or is it that Brownian diffusion facilitates stress propagation? Any evidence in the paper what actually the case is?"

RESPONSE:

We have now clarified what we mean: with $\lambda=0$, the density simply obeys a diffusion equation with $G/\hat{\gamma}$ the diffusion coefficient. As a simple constitutive relation has the stress proportional to ρ (as discussed in the text), within this limit also the stress will obey a diffusion equation. We additionally gave a reference to Ref. [16] where a similar diffusive law was derived for compressible (G not infinite) colonies, with Hertzian (rather than Hookean) elasticity.

COMMENT:

"12) The section "Theoretical description" reads extremely vague and speculative, compared to what the authors claim in their abstract. It is highly unclear where do the reported length scales come from? Are they general to active nematics or do they represent a special case to bacterial colonies? Why, according to the authors, does the active stress depend only on the elastic modulus of the cell envelope and not, for instance, on the growth rate? How do the authors account for the extra cellular matrix - will this not contribute to the elasticity of the cell modulus? How do the authors account for this? Importantly, the referee completely misses any mapping in this regard, between the theoretical description and the experiments that they report."

RESPONSE:

As we have now added the discussion of direct numerical simulations of a full theory based on the elements of our previous schematic theory, these comments no longer apply as the new theory renders the comparison between experiments and simulations/theory both clearer and more quantitative.

For completeness we nevertheless reply to each of the reviewer's questions below:

(i) for the length scales: (a) l_a is the domain size, and now we have discussed this in more detail (for both wet systems and dry ones, the case relevant to our colonies), (b) l_σ is the classic length scale which appears in a diffusion equation (here with diffusivity $G/\hat{\gamma}$) and a reaction rate (here the growth rate);

(ii) l_a is general to dry active nematics; l_σ requires compressibility and growth, so is unique to our current system;

(iii) elasticity and active stress may be thought to implicitly depend on growth rate: they would in our theory as well, through their dependence on cell density;

(iv) the extracellular matrix may play a role but a numerical renormalisation of K and ζ would not change things qualitatively; it provides an extra level of detail which is, for instance, never included in standard number-conserving active matter theories, see e.g. M. C. Marchetti et al, Rev. Mod. Phys. 85, 1143 (2013).

REVIEWERS' COMMENTS:

Reviewer #1 (Remarks to the Author):

The authors have done a satisfactory job in responding to my concerns and improving the overall flow of the manuscript. I can now recommend publishing this work in Nature Communications.

Reviewer #2 (Remarks to the Author):

My main criticism was the need to supplement the paper with a more extended account of theory and simulations to better interpret the reported results. Authors have made an effort in this respect and I appreciate it.

Also, some minor points I raised before have been revised and clarified in the new version.

I do not have major objections towards the publication of the paper

Reviewer #3 (Remarks to the Author):

The authors have addressed mostly all my comments.

I am still not 100% convinced about the fact that the authors have found strong evidence for a linear scaling Fig. 2 (c).

To me it is more like a consistency (the authors may wanna phrase it like that) but to be fair, the authors do not claim that the linear scaling is clearly visible or so, but to make the analogy to the Hubble expansion they need it ...)

All remaining edits are great and I appreciate the hard work of the authors.

Overall, good works that deserves publication in my eyes.

Reply to Reviewers:

We thank the reviewer for their careful reading of our manuscript, as well as for their positive comments and for recommending our work for publication.

There is only a minor remaining comment from Reviewer #3 that we dealt with as follows.

COMMENT:

"I am still not 100% convinced about the fact that the authors have found strong evidence for a linear scaling Fig. 2 (c). To me it is more like a consistency (the authors may wanna phrase it like that) but to be fair, the authors do not claim that the linear scaling is clearly visible or so, but to make the analogy to the Hubble expansion they need it...)"

RESPONSE:

We have now followed the Reviewer's advice and mentioned more clearly that the results in Fig. 2(c) are consistent with the linear scaling expected theoretically, rather than a demonstration of it (see new modified second paragraph in the section called "'Hubble' expansion").